# DTX3L ubiquitin ligase ubiquitinates single-stranded nucleic acids

Emily L Dearlove[1,2†], Chatrin Chatrin[1,2†‡], Lori Buetow[1], Syed F Ahmed[1], Tobias Schmidt[1], Martin Bushell[1,2], Brian O Smith[3], Danny T Huang[1,2]*

[1]Cancer Research UK Scotland Institute, Garscube Estate, Switchback Road, Glasgow, United Kingdom; [2]School of Cancer Sciences, University of Glasgow, Glasgow, United Kingdom; [3]School of Molecular Biosciences, University of Glasgow, Glasgow, United Kingdom

**\*For correspondence:**
d.huang@crukscotlandinstitute.
ac.uk

[†]These authors contributed equally to this work

**Present address:** [‡]Sir William Dunn School of Pathology, University of Oxford, Oxford, United Kingdom

**Abstract** Ubiquitination typically involves covalent linking of ubiquitin (Ub) to a lysine residue on a protein substrate. Recently, new facets of this process have emerged, including Ub modification of non-proteinaceous substrates like ADP-ribose by the DELTEX E3 ligase family. Here, we show that the DELTEX family member DTX3L expands this non-proteinaceous substrate repertoire to include single-stranded DNA and RNA. Although the N-terminal region of DTX3L contains single-stranded nucleic acid binding domains and motifs, the minimal catalytically competent fragment comprises the C-terminal RING and DTC domains (RD). DTX3L-RD catalyses ubiquitination of the 3'-end of single-stranded DNA and RNA, as well as double-stranded DNA with a 3' overhang of two or more nucleotides. This modification is reversibly cleaved by deubiquitinases. NMR and biochemical analyses reveal that the DTC domain binds single-stranded DNA and facilitates the catalysis of Ub transfer from RING-bound E2-conjugated Ub. Our study unveils the direct ubiquitination of nucleic acids by DTX3L, laying the groundwork for understanding its functional implications.

## eLife assessment

This **important** study reports the discovery of a novel nucleotide ubiquitylation activity by the DTX3L E3 ligase. **Solid** evidence is presented for ubiquitin attachment to single-stranded oligonucleotides. This very interesting biochemical finding can be used as a starting point for studies to establish relevance in a physiological setting.

## Introduction

Ubiquitination—the covalent attachment of Ub to a substrate—is a versatile post-translational modification involved in the regulation of various cellular functions (*Yau and Rape, 2016*). The most widely studied examples of this modification involve the conjugation of Ub to lysine residues in proteinaceous substrates. Non-lysine ubiquitination of protein substrates via serine (*Wang et al., 2009*) and threonine (*Shimizu et al., 2010*) residues has also been demonstrated. The modification of these residues produces an ester linkage in contrast to the canonical isopeptide amide bond that is observed in the ubiquitination of lysine residues (*Wang et al., 2007*). In recent years, non-proteinaceous ubiquitination substrates have emerged. These substrates include bacterial lipopolysaccharide which is ubiquitinated by RNF213 in the immune response to *Salmonella* (*Otten et al., 2021*), glucosaccharides which are ubiquitinated by HOIL-1 (*Kelsall et al., 2022*) and adenosine 5'-diphosphate (ADP)–ribose (ADPr) which is ubiquitinated by the DELTEX family of ubiquitin ligases (E3s) (*Chatrin et al., 2020*; *Yang et al., 2017*; *Zhu et al., 2022*). Collectively, these studies have uncovered that the diversity of

substrates undergoing ubiquitination expands beyond that of proteinaceous substrates. Additionally, they highlight that this mechanism is not limited to one E3 ligase family.

The DELTEX (DTX) E3 family is comprised of five RING E3s, DTX1, DTX2, DTX3, DTX3L, and DTX4, which share a conserved C-terminal region containing the RING and DELTEX C-terminal (DTC) domains (*Chatrin et al., 2020*; *Ahmed et al., 2020*). Whilst the RING domain functions by recruiting E2 thio-esterified with Ub (E2~Ub) to catalyse Ub transfer to a substrate (*Deshaies and Joazeiro, 2009*), the DTC domain, which is connected to the RING domain through a short flexible linker remained, until recently, without a reported function. DTX3L, which forms a heterodimer with poly(ADP-ribose) polymerase 9 (PARP9) (*Takeyama et al., 2003*), was initially proposed to be an activator of PARP9 ADP-ribosyltransferase activity, with the complex being shown to catalyse ADP-ribosylation of Ub in the presence of nicotinamide adenine dinucleotide (NAD$^+$) and ubiquitination components including E1, E2, Ub, Mg$^{2+}$, and ATP (*Yang et al., 2017*). It was originally proposed that PARP9 was responsible for this activity, despite previous reports of its catalytic inactivity (*Vyas et al., 2014*). When we tested DTX3L independently of PARP9, it was sufficient to catalyse ADP-ribosylation of the C-terminus of Ub (*Chatrin et al., 2020*). Notably, the C-terminal RING and DTC domains (hereafter referred to as RD domains) are the minimal fragments required to catalyse this reaction and all members of the DTX family share this capability (*Chatrin et al., 2020*; *Yang et al., 2017*; *Zhu et al., 2022*). Structural analyses of RD domains revealed that the DTC domain contains a pocket that binds ADPr and NAD$^+$ (*Chatrin et al., 2020*; *Ahmed et al., 2020*). This enables RD domains to recruit ADPr-modified substrates and catalyse their ubiquitination in a poly-ADP-ribose (PAR)-dependent manner (*Ahmed et al., 2020*). In addition, in the presence of NAD$^+$ and ubiquitination components, RD domains catalyse ADP-ribosylation of Ub (*Chatrin et al., 2020*). The chemical linkage of ADP-ribosylated Ub was recently discovered to occur between the carboxylate group of Ub Gly$^{76}$ and the 3'-hydroxyl group of the adenosine-proximal ribose of ADPr or NAD$^+$, highlighting Ub modification of ADPr or NAD$^+$ rather than the canonical ADP-ribosylation of substrate that involves the release of nicotinamide from NAD$^+$ followed by attachment of ADPr via the C1 atom of the nicotinamide ribose (*Zhu et al., 2022*). Furthermore, recent studies have demonstrated that the RD domains of both DTX2 and DTX3L can ubiquitinate the 3'-hydroxyl group of the adenosine-proximal ribose of free ADPr as well as ADPr moieties present on ADP-ribosylated proteins (*Zhu et al., 2022*) and ADP-ribosylated nucleic acids (*Zhu et al., 2024*).

DTX3L has a unique N-terminus lacking the WWE domains and proline-rich regions found in the other DTX family members. The development of the AlphaFold Protein Structure Database (*Jumper et al., 2021*) has allowed the prediction of unsolved protein structures to a higher degree of confidence than previously possible. For DTX3L, this allowed the generation of a model of the protein that revealed putative domains in the N-terminal region. When queried in the DALI server, which enables protein structures to be compared in 3D, these domains were found to be structurally similar to K Homology (KH) domains and RNA recognition motifs (RRMs) which bind single-stranded DNA (ssDNA) and RNA (ssRNA). Here, we showed that DTX3L was able to bind a variety of sequences of ssDNA and ssRNA. Unexpectedly, we discovered that DTX3L was able to ubiquitinate ssDNA and ssRNA. Biochemical and structural analyses established that the DTX3L-RD domain binds single-stranded nucleic acids (ssNAs) and catalyses the ubiquitination of ssNAs. These findings present a previously unidentified direct Ub modification of nucleic acids.

## Results
### DTX3L binds and ubiquitinates ssRNA and ssDNA

To gain insight into the function of DTX3L, we took the AlphaFold-predicted structural domains of DTX3L (*Figure 1A*) and conducted a search on the DALI server to compare the proposed 3D domains of DTX3L to structures in the Protein Data Bank (PDB). Several of these domains share structural similarities to KH domains (*Figure 1B and C*) with the N-terminal domain resembling an RRM. KH domains, which bind both ssRNA (*Baber et al., 1999*) and ssDNA (*Baber et al., 2000*), comprise a three-stranded anti-parallel β-sheet packed alongside three α-helices whereas RRMs are composed of two α-helices and four anti-parallel strands packed together in a β-α-β-β-α-β fold (*Cléry et al., 2008*). To determine if DTX3L binds ssRNA or ssDNA, we used ssDNA and ssRNA oligonucleotides labelled with 6-FAM on the 5'-end (D1-D9 and R1-R9, respectively; *Table 1*; *Figure 1—figure supplement*

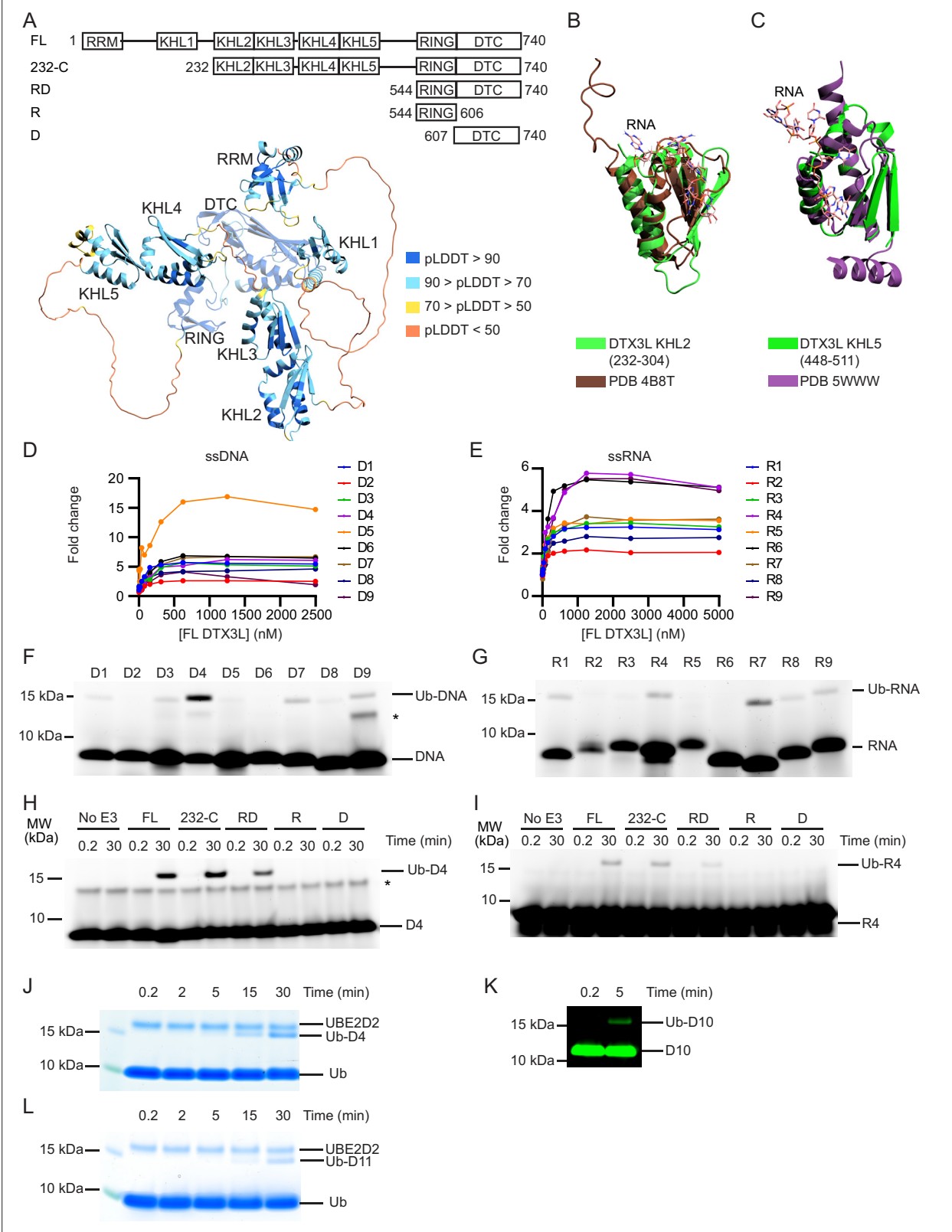

**Figure 1.** DTX3L catalyses the ubiquitination of single-stranded nucleic acids. (**A**) Cartoon representation of the AlphaFold model of DTX3L. Domains are coloured according to model confidence. Domain architecture of DTX3L constructs is shown above the model. (**B**) DTX3L KHL2 (232–304) prediction shown in green overlaid with the third K Homology (KH) domain of KSRP in complex with AGGGU RNA sequence (PDB 4B8T) shown in brown. (**C**) DTX3L KHL5 (448–511) prediction shown in green overlaid with the MEX-3C KH1 domain in complex with GUUUAG RNA sequence (PDB 5WWW)

*Figure 1 continued on next page*

*Figure 1 continued*

shown in purple. (**D**) Fold change of fluorescence polarisation of 6-FAM-labelled ssDNA D1-9 upon titrating with full-length DTX3L. (**E**) As in (**D**) but with 6-FAM-labelled ssRNA R1-9. Data points for (**D**) and (**E**) are shown in *Figure 1—source data 1 and 2*, respectively. (**F**) Fluorescently detected SDS-PAGE gel of in vitro ubiquitination of 6-FAM-labelled ssDNA D1-9 oligonucleotides by FL DTX3L in the presence of E1, UBE2D2, Ub, Mg$^{2+}$-ATP. (**G**) As in (**F**) but with 6-FAM-labelled ssRNA R1-9 oligonucleotides. (**H**) Fluorescently detected SDS-PAGE gel of in vitro ubiquitination of 6-FAM-labelled ssDNA D4 by DTX3L variants (FL, full length; RD, RING-DTC domains; R, RING domain; D, DTC domain) in the presence of E1, UBE2D2, Ub, Mg$^{2+}$-ATP. (**I**) As in (**H**) but with 6-FAM-labelled ssRNA R4. (**J**) Coomassie-stained SDS-PAGE gel of in vitro ubiquitination of 6-FAM-labelled ssDNA D4 by DTX3L-RD in the presence of E1, UBE2D2, Ub, and Mg$^{2+}$-ATP. (**K**) Fluorescently detected SDS-PAGE gel of in vitro ubiquitination 5′ IRDye 800 ssDNA D10 by DTX3L-RD in the presence of E1, UBE2D2, Ub, Mg$^{2+}$-ATP. (**L**) Coomassie-stained SDS-PAGE gel of in vitro ubiquitination of ssDNA D11 by DTX3L-RD in the presence of E1, UBE2D2, Ub, and Mg$^{2+}$-ATP. Asterisks in (**F**) and (**H**) indicate contaminant bands from single-stranded DNA (ssDNA) or single-stranded RNA (ssRNA). Raw unedited and uncropped gel images for (F–L) are shown in *Figure 1—source data 3 and 4*, respectively.

The online version of this article includes the following source data and figure supplement(s) for figure 1:

**Source data 1.** Fluorescence polarisation data related to *Figure 1D*.

**Source data 2.** Fluorescence polarisation data related to *Figure 1E*.

**Source data 3.** Raw unedited gels for *Figure 1*.

**Source data 4.** Uncropped and labelled gels for *Figure 1*.

**Figure supplement 1.** DTX3L binds and ubiquitinates single-stranded nucleic acids (ssNAs).

**Figure supplement 1—source data 1.** Fluorescence polarisation data related to *Figure 1—figure supplement 1B*.

**Figure supplement 1—source data 2.** Fluorescence polarisation data related to *Figure 1—figure supplement 1C*.

**Figure supplement 1—source data 3.** Fluorescence polarisation data related to *Figure 1—figure supplement 1D*.

**Figure supplement 1—source data 4.** Fluorescence polarisation data related to *Figure 1—figure supplement 1E*.

**Figure supplement 1—source data 5.** Raw unedited gels for *Figure 1—figure supplement 1*.

**Figure supplement 1—source data 6.** Uncropped and labelled gels for *Figure 1—figure supplement 1*.

*1A*) and either full-length (FL) DTX3L or a truncated 232-C construct containing the two tandem KH-like (KHL) domains and C-terminal RING-DTC that is competent in binding PARP9 (*Ashok et al., 2022*). Both DTX3L constructs bound to all tested sequences of ssRNA and ssDNA, inducing at least a twofold change in polarisation (*Figure 1D and E*; *Figure 1—figure supplement 1B, C*). Likewise, at least a twofold change in polarisation was observed when DTX3L-PARP9 complex binding was tested with these nucleic acids, demonstrating that PARP9 does not occlude ssDNA or ssRNA binding by DTX3L (*Figure 1—figure supplement 1D, E*).

Based on the structural similarity between ADPr and nucleic acids (*Figure 1—figure supplement 1F*), we hypothesised that RNA and DNA could also be ubiquitinated by DTX3L. To investigate, we performed in vitro ubiquitination assays of FAM-labelled ssDNA or ssRNA by DTX3L with the following components: E1, UBE2D2, Ub, and Mg$^{2+}$-ATP. Ubiquitination of FAM-labelled ssDNA and ssRNA was detected by SDS-PAGE using a fluorescent scanner (Typhoon FLA7000). The appearance of an ~15 kDa band closely matched the combined molecular weights of Ub (8564 Da) and 20 nt FAM-labelled single-stranded nucleic acid (~6900–7300 Da), suggesting that both ssDNA (*Figure 1F*) and ssRNA (*Figure 1G*) were ubiquitinated by DTX3L. Due to the high signal-to-noise ratio observed with D4 compared to other oligonucleotides, this sequence was used for subsequent experiments. Because R4 is the corresponding sequence of ssRNA this was also taken forward.

The minimal fragment of DTX3L required for Ub-ADPr formation is the conserved C-terminal RING-DTC domain (*Chatrin et al., 2020*). To investigate the minimal fragment of DTX3L necessary for the ubiquitination of ssNAs, we tracked the formation of ubiquitinated ssDNA (Ub-DNA) or ssRNA (Ub-RNA) by DTX3L 232-C, RD, RING, or DTC. DTX3L 232-C lacks the N-terminal RRM and KHL1 domains, where KHL1 domain was recently proposed to enable oligomerisation (*Vela-Rodríguez et al., 2024*). While DTX3L 232-C exhibited reduced autoubiquitination compared to DTX3L FL (*Figure 1—figure supplement 1G*), the rates of UBE2D2~Ub discharge were comparable (*Figure 1—figure supplement 1H*). This suggests that DTX3L 232-C is as catalytically competent as DTX3L FL, but lacks accessible lysine sites necessary for autoubiquitination. DTX3L 232-C exhibited similar activity as DTX3L FL in catalysing the formation of Ub-DNA (*Figure 1H*) and Ub-RNA (*Figure 1I*). DTX3L-RD also catalysed the formation of Ub-DNA (*Figure 1H*) and Ub-RNA (*Figure 1I*), whereas neither the RING (R) nor the DTC (D) alone sufficed. Hints of the ~15 kDa Ub-DNA band were also observed on the corresponding Coomassie-stained SDS-PAGE gel (*Figure 1—figure supplement 1I*, arrows).

**Table 1.** List of nucleotide sequences used in this study.

| Name | Type | Sequence | Modifications |
|------|------|----------|---------------|
| D1 | ssDNA | 5' TGTTTGTTTGTTTGTTTGTT 3' | 5' FAM |
| D2 | ssDNA | 5' GCGCGCGCGCGCGCGCGCGC 3' | 5' FAM |
| D3 | ssDNA | 5' AGTGAGTGAGTGAGTGAGTG 3' | 5' FAM |
| D4 | ssDNA | 5' CAACAACAACAACAACAACA 3' | 5' FAM |
| D5 | ssDNA | 5' AGAGAGAGAGAGAGAGAGAG 3' | 5' FAM |
| D6 | ssDNA | 5' TCTCTCTCTCTCTCTCTCTC 3' | 5' FAM |
| D7 | ssDNA | 5' TTTTTTTTTTTTTTTTTTTT 3' | 5' FAM |
| D8 | ssDNA | 5' GTGCTGCGCTGCGCTGTGCT 3' | 5' FAM |
| D9 | ssDNA | 5' AAAAAAAAAAAAAAAAAAAA 3' | 5' FAM |
| D10 | ssDNA | 5' CAACAACAACAACAACAACA 3' | 5' IRDye 800 |
| D11 | ssDNA | 5' CAACAACAACAACAACAACA 3' | None |
| D12 | dsDNA | 5' CAACAACAACAACAACAACA 3' + 3' TGTTGTTGTTGTTGTTGTTG 5' | 5' FAM None |
| D13 | ssDNA | 5' CAACAACAACAACAACAACA 3' | 3' FAM |
| D14 | ssDNA | 5' CAACAACAACAACAACAACA 3' | 5' IRD800 3' 2FA |
| D15 | ssDNA | 5' CAACAACAACAACAACAACA 3' | 5' IRDye 800 3' Phos |
| D16 | ssDNA | 5' CAACAACAACAACAACAACT 3' | 5' FAM |
| D17 | ssDNA | 5' CAACAACAACAACAACAACG 3' | 5' FAM |
| D18 | ssDNA | 5' CAACAACAACAACAACAACC 3' | 5' FAM |
| D19 | ssDNA | 5' CAACAACAACAACAACAAAA 3' | 5' FAM |
| D20 | ssDNA | 5' CAACAACAACAACAACAATA 3' | 5' FAM |
| D21 | ssDNA | 5' CAACAACAACAACAACAAGA 3' | 5' FAM |
| D22 | ssDNA | 5' CAACA 3' | 5' FAM |
| D23 | ssDNA | 5' AACAACAACA 3' | 5' FAM |
| D24 | ssDNA | 5' AACAACAACAACAACAACAACAAC AACAACAACAACA 3' | 5' FAM |
| D25 | ssDNA | 5' CAACAACAACAACAACAACAACAACAAC AACAACAACAACAACAACAACAACAAC AACAACAACAACAACAACA 3' | 5' FAM |
| D26 | ssDNA | CGGCACATCACTCTTCAACA | 5' FAM |
| D27 | dsDNA | 5' CGGCACATCACTCTTCAACA 3' + 5' GTTGAAGAGTGATGTGCCG 3' | 5' FAM None |
| D28 | dsDNA | 5' CGGCACATCACTCTTCAACA 3' + 5' TTGAAGAGTGATGTGCCG 3' | 5' FAM None |
| D29 | dsDNA | 5' CGGCACATCACTCTTCAACA 3' + 5' TGAAGAGTGATGTGCCG 3' | 5' FAM None |
| D30 | ssDNA | CAACA | None |
| D31 | ssDNA | 5' CAACAACAACAACAACAACA 3' | 3' Phos |
| R1 | ssRNA | 5' UGUUUGUUUGUUUGUUUGUU 3' | 5' FAM |
| R2 | ssRNA | 5' GCGCGCGCGCGCGCGCGCGC 3' | 5' FAM |
| R3 | ssRNA | 5' AGUGAGUGAGUGAGUGAGUG 3' | 5' FAM |
| R4 | ssRNA | 5' CAACAACAACAACAACAACA 3' | 5' FAM |

*Table 1 continued on next page*

Table 1 continued

| Name | Type | Sequence | Modifications |
|------|------|----------|---------------|
| R5 | ssRNA | 5' AGAGAGAGAGAGAGAGAGAG 3' | 5' FAM |
| R6 | ssRNA | 5' UCUCUCUCUCUCUCUCUCUC 3' | 5' FAM |
| R7 | ssRNA | 5' UUUUUUUUUUUUUUUUUUUU 3' | 5' FAM |
| R8 | ssRNA | 5' GUGCUGCGCUGCGCUGUGCU 3' | 5' FAM |
| R9 | ssRNA | 5' AAAAAAAAAAAAAAAAAAAA 3' | 5' FAM |

We took advantage of the Coomassie staining method of detection for Ub-DNA, along with an increased concentration of D4, to confirm this and demonstrate that the formation of Ub-DNA is time-dependent (*Figure 1J*). To validate that Ub modification did not occur on the FAM label, we tested ssDNA labelled with IRDye 800 (D10, *Table 1*; *Figure 1—figure supplement 1J*) and unlabelled ssDNA (D11, *Table 1*) in a ubiquitination assay. An ~15 kDa band corresponding to the predicted molecular weight of Ub-DNA was also observed using ssDNA with the alternative label (*Figure 1K*) and unlabelled ssDNA (*Figure 1L*). Together, our data demonstrate that DTX3L binds and ubiquitinates ssNAs and that DTX3L-RD is the minimal fragment required to catalyse this reaction.

## The 3'-end of ssDNA is modified on its 3' hydroxyl

Based on the finding that the RD was sufficient to catalyse the ubiquitination of ssDNA D4, we confirmed that this fragment was able to ubiquitinate the same ssDNAs as FL DTX3L (*Figure 1—figure supplement 1K*). Hence, this fragment was used to probe the additional requirements of the reaction. Initially, to confirm that product formation was dependent on the ubiquitination process, we removed each reactant individually. Formation of Ub-DNA or Ub-RNA by DTX3L-RD was only detected when all components of the ubiquitination cascade (E1, UBE2D2, Ub, $Mg^{2+}$-ATP) were present (*Figure 2A and B*). To validate that ssDNA and ssRNA were modified with Ub and not another component in the reaction, we treated the resulting product with USP2, a promiscuous deubiquitinating enzyme. The ~15 kDa band disappeared upon treatment with USP2, which is consistent with the removal of Ub from ssDNA (*Figure 2C*) and ssRNA (*Figure 2D*) and shows that this modification is reversible. Treatment of Ub-DNA and Ub-RNA with Benzonase, an endonuclease which degrades DNA and RNA, caused the disappearance of both the Ub-DNA and DNA bands (*Figure 2C*) and Ub-RNA and RNA bands (*Figure 2D*). Poly(ADP-ribose) glycohydrolase (PARG), an ADPr hydrolase that cleaves ADPr-linked to substrate, was recently shown to cleave Ub-ADPr attached to a nucleic acid substrates (*Zhu et al., 2024*). Treatment with PARG had no effect on the Ub-DNA band (*Figure 2C*) or Ub-RNA band (*Figure 2D*), thereby demonstrating that this reaction is distinct from the attachment of Ub to the ADPr moiety on ADP-ribosylated nucleic acids (*Zhu et al., 2024*). These findings showed that the 15 kDa product was ubiquitinated ssNAs.

Our experiments suggested a common mechanism of modification between ssRNA and ssDNA; therefore, we focused our subsequent experiments on the most readily detectable substrate, ssDNA D4. To ascertain whether this reaction was specific to ssDNA, we followed the formation of Ub-DNA with 6-FAM-labelled dsDNA and DTX3L-RD. Based on the finding that the RD fragment was sufficient to catalyse the ubiquitination of ssDNA, this fragment was used to probe the additional requirements of the reaction. DTX3L-RD did not ubiquitinate dsDNA (D12, *Table 1*; *Figure 2E*) suggesting that the protein is specific for ssDNA. We then compared ubiquitination of ssDNA labelled with 6-FAM on the 5'-end or 3'-end (D13, *Table 1*; *Figure 1—figure supplement 1A*, L) and found that the 6-FAM label on the 3'-end blocked ubiquitination of ssDNA (*Figure 2F*). Based on this finding we hypothesised that the 3'-end of the oligonucleotide is (1) the site of modification or (2) important for oligonucleotide binding. We next investigated the chemical nature of the covalent bond between Ub and ssDNA. To see if the bond was hydrolysed under basic conditions, we treated our reaction with pH 9.5 buffer which led to the disappearance of the Ub-DNA band over time (*Figure 2G*). Additionally, treatment with $NH_2OH$ resulted in the rapid disappearance of the Ub-DNA band (*Figure 2H*). Together, this suggested that the linkage is likely to be an ester, and that the modification occurs on a ribose moiety, leaving only the 3' ribose as a viable modification site. To validate this hypothesis, we designed an

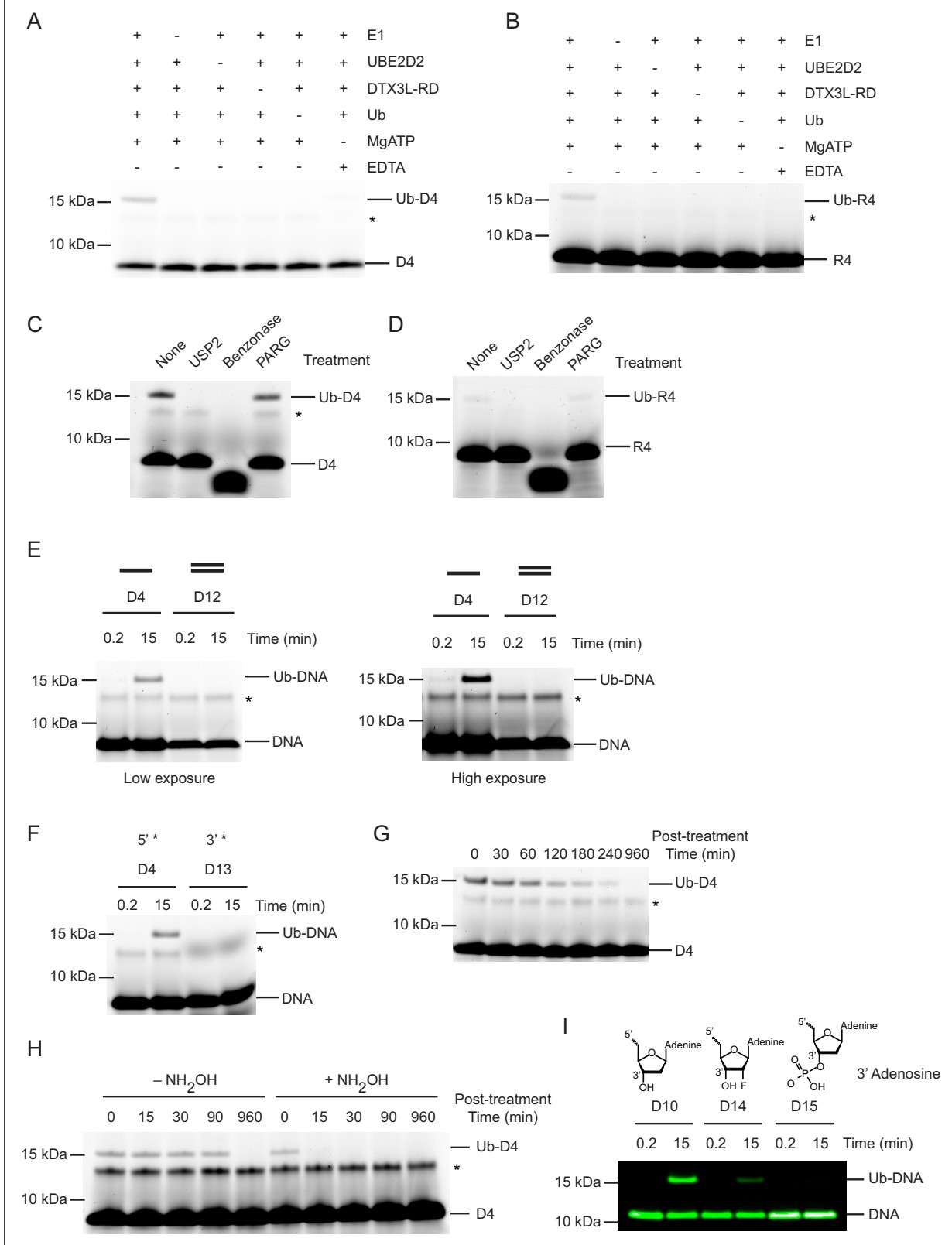

**Figure 2.** Ubiquitin (Ub) modification of single-stranded DNA (ssDNA) occurs at the 3' hydroxyl at the 3' end. (**A**) Fluorescently detected SDS-PAGE gel of in vitro ubiquitination of ssDNA D4 in which E1, UBE2D2, DTX3L-RD, Ub, or $Mg^{2+}$-ATP have been omitted. (**B**) As in (**A**) but with 6-FAM-labelled single-stranded RNA (ssRNA) R4. (**C**) Fluorescently detected SDS-PAGE gel of in vitro ubiquitination of 6-FAM-labelled ssDNA D4 by DTX3L-RD in the presence of E1, UBE2D2, Ub, $Mg^{2+}$-ATP subsequently treated with USP2, Benzonase, Poly(ADP-ribose) glycohydrolase (PARG) or not treated (None). (**D**)

*Figure 2 continued on next page*

*Figure 2 continued*

As in (**C**) but with 6-FAM-labelled ssRNA R4. (**E**) Fluorescently detected SDS-PAGE gel of in vitro ubiquitination of 6-FAM-labelled ssDNA D4 and dsDNA D12 oligonucleotides by DTX3L-RD (left panel) and at increased exposure (right panel) in the presence of E1, UBE2D2, Ub, $Mg^{2+}$-ATP. (**F**) As in (**E**) but with 6-FAM-labelled ssDNA D4 (5' label) and D13 (3' label) oligonucleotides. (**G**) Fluorescently detected SDS-PAGE gel of in vitro ubiquitinated 6-FAM-labelled ssDNA D4 subsequently treated with pH 9.5 buffer for the times indicated. (**H**) Fluorescently detected SDS-PAGE gel of in vitro ubiquitinated 6-FAM-labelled ssDNA D4 subsequently treated with 1.5 M $NH_2OH$ at pH 9 for the times indicated. (**I**) As in (**E**) but with 5' IRDye 800 ssDNA D10, D14 and D15 oligonucleotides. Asterisks in (**A–C**) and (**E–H**) indicate contaminant bands from ssDNA or ssRNA. Raw unedited and uncropped gel images are shown in *Figure 2—source data 1 and 2*, respectively.

The online version of this article includes the following source data for figure 2:

**Source data 1.** Raw unedited gels for *Figure 2*.

**Source data 2.** Uncropped and labelled gels for *Figure 2*.

array of 5'-FAM labelled ssDNAs in which various positions on the ribose ring of the 3'-end nucleotide were modified and tested their ability to act as a substrate for ubiquitination by DTX3L-RD. Whilst ssDNA with a fluorine atom attached at the 2' position of the ribose ring (D14) was ubiquitinated, the addition of a phosphate moiety to the 3' hydroxyl of the ribose ring (D15) ablated the formation of Ub-DNA, thereby confirming this is the site of modification (D10, D14-D15, *Table 1*; *Figure 2I*). This is consistent with the site of Ub modification of ADPr where Ub is also attached to the 3' hydroxyl of the adenine-proximal ribose (*Zhu et al., 2022*). These data support that Ub modification of ssDNA occurs at the 3' hydroxyl of the 3' nucleotide.

## Nucleotide sequence requirements for Ub-DNA formation

How the sequence and length of ssDNA affect Ub-DNA formation is unclear. To investigate, we first probed the base preference at the last or penultimate position by varying the nucleic acid to A, T, C, or G of oligonucleotide D4. Whilst only the sequences ending with A and to some extent with G could be ubiquitinated by DTX3L-RD, the requirement for the penultimate nucleotide was less strict, with sequences ending CA, TA, and AA all being ubiquitinated (D16-D21, *Table 1*; *Figure 3A and B*). Thus far we had only tested sequences of 20 nucleotides in length; therefore, we next took the D4 sequence and generated four ssDNA nucleotides with this same CAA repeat unit and ranging in length from five to 80 nucleotides (D22-D25, *Table 1*). When tested in our assays, all four of these nucleotide substrates were ubiquitinated (*Figure 3C and D*). Despite dsDNA being unable to act as a substrate for ubiquitination, our discovery that ubiquitination occurs on the 3' nucleotide of ssDNA prompted us to test the hypothesis that dsDNA with a 3' overhang could be ubiquitinated. Whilst dsDNA with a single 3' nucleotide overhang could not be ubiquitinated, dsDNA with a two or three 3' nucleotide overhang could be modified with Ub (D26-D29, *Table 1*; *Figure 3E*). Our data show that DTX3L modification of DNA requires that the two nucleotides at the 3' end be single-stranded. Overall, these data demonstrate the ability of DTX3L-RD to ubiquitinate ssDNA of varying lengths and specific 3' dinucleotides.

## ssNA and ADPr share a binding site on DTX3L's DTC domain

Because the modification of ssDNA and ADPr occurs at the same position on the nucleoside, we next explored the possibility that they share the same binding site on DTX3L-RD. ADPr binds the DTC domain of DTX2 (*Ahmed et al., 2020*), a region which is conserved in the DTX family. Therefore, we purified $^{15}N$-DTX3L-RD and acquired $^{1}H$-$^{15}N$ heteronuclear single-quantum coherence (HSQC) spectra of $^{15}N$-DTX3L-RD alone and in the presence of ADPr or unlabelled ssDNA. We utilised the unlabelled version of the 5-mer oligonucleotide ssDNA (D30, *Table 1*) for NMR analysis, which was sufficient to produce the Ub-DNA product (*Figure 3C*). Upon titration of ADPr or 5-nucleotide ssDNA, chemical shift perturbations (CSPs) occurred, indicating DTX3L-RD binds both ADPr and ssDNA (*Figure 4A*; *Figure 4—figure supplement 1A, B*). The similarity in the CSPs indicates that there is some overlap in the binding sites of ADPr and ssDNA (*Figure 4A–C*). To further verify that there is conservation between the binding sites of these two substrates, excess ADPr was added to the ubiquitination reaction components along with DTX3L-RD and ssDNA D4. The presence of ADPr inhibited the formation of Ub-DNA and Ub-RNA (*Figure 4D* and *Figure 4—figure supplement 1C*), suggesting they share the same binding site. We next added excess ssDNA D31, a ssDNA with the same sequence as D4 but with a phosphate moiety attached to the 3' hydroxyl, to the ubiquitination reaction components

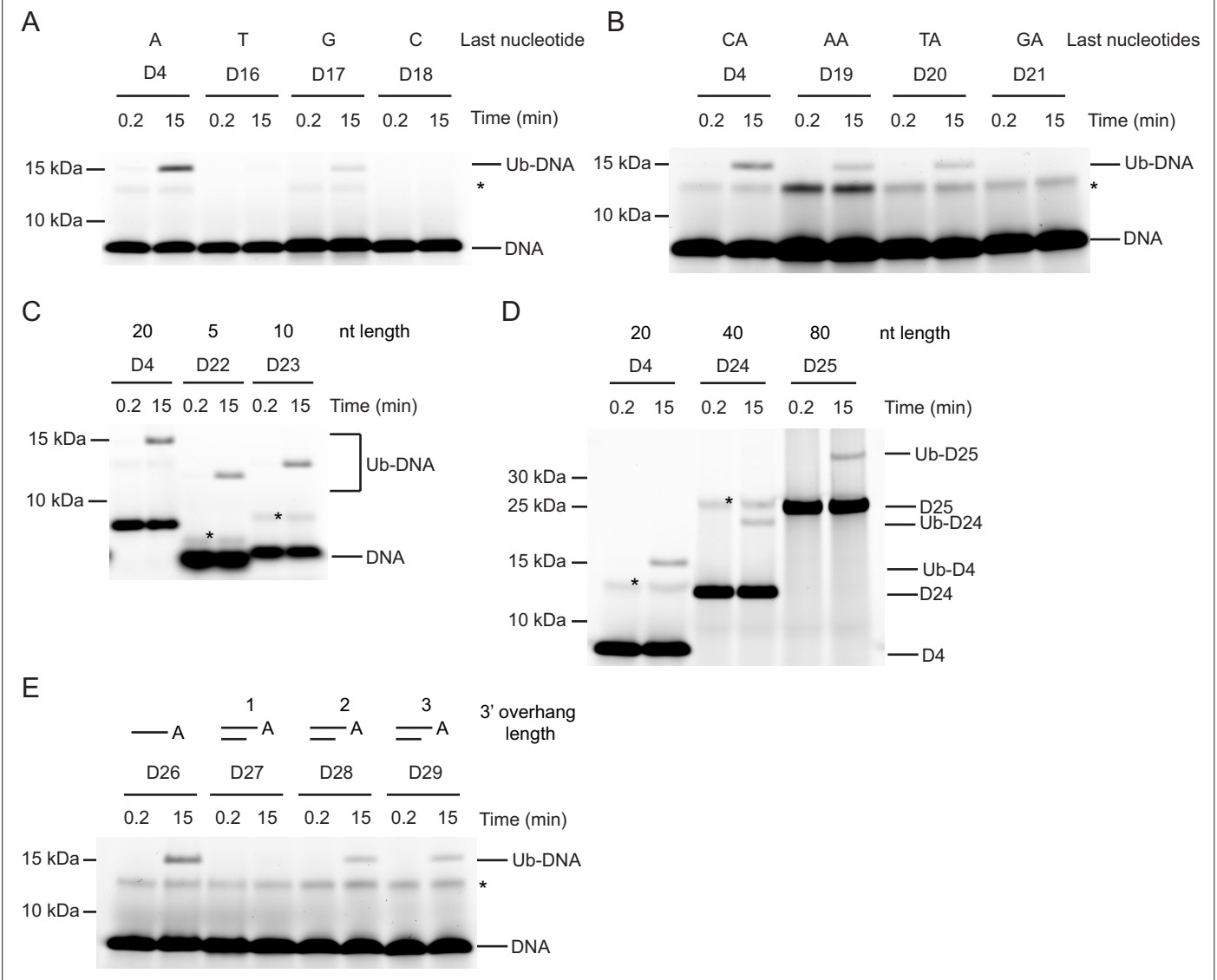

**Figure 3.** Nucleotide sequence requirements for ubiquitin (Ub)-DNA formation. Fluorescently detected SDS-PAGE gel of in vitro ubiquitination (in the presence of E1, UBE2D2, Ub, and $Mg^{2+}$-ATP) of (**A**) 6-FAM-labelled single-stranded DNA (ssDNA) D4, D16, D17 and D18 by DTX3L-RD. (**B**) 6-FAM-labelled ssDNA D4, D19, D20, and D21 by DTX3L-RD. (**C**) 6-FAM-labelled ssDNA D4, D22, and D23 by DTX3L-RD. (**D**) 6-FAM-labelled ssDNA D4, D24 and D25 by DTX3L-RD. (**E**) 6-FAM-labelled ssDNA D26, dsDNA D27, D28 and D29 by DTX3L-RD. Asterisks indicate contaminant bands from ssDNA. Raw unedited and uncropped gel images are shown in *Figure 3—source data 1 and 2*, respectively.

The online version of this article includes the following source data for figure 3:

**Source data 1.** Raw unedited gels for *Figure 3*.

**Source data 2.** Uncropped and labelled gels for *Figure 3*.

along with DTX3L-RD and biotin-$NAD^+$. As labelled ADPr is not readily available, we used biotin-$NAD^+$ to monitor Ub-biotin-$NAD^+$ formation. The addition of D31 inhibited the formation of Ub-$NAD^+$ (*Figure 4E*). Because D31 has a phosphate moiety blocking the 3' hydroxyl ubiquitination site, it cannot be ubiquitinated; therefore, it must be directly competing for binding.

To assess DTX3L-RD preference for nucleic acid or ADPr, we performed a kinetic analysis of Ub-D4 and Ub-F-$NAD^+$ formation by DTX3L-RD using D4 and 6-Fluo-10-$NAD^+$ (F-$NAD^+$), respectively, as substrates. DTX3L-RD displayed a $k_{cat}$ value of 0.0358±0.0034 $min^{-1}$ and a $K_m$ value of 6.56±1.80 µM for Ub-D4 formation, whereas the Michaelis-Menten curve did not reach saturation for Ub-F-$NAD^+$ formation (*Figure 4F* and *Figure 4—figure supplement 1D–G*). Comparison of the

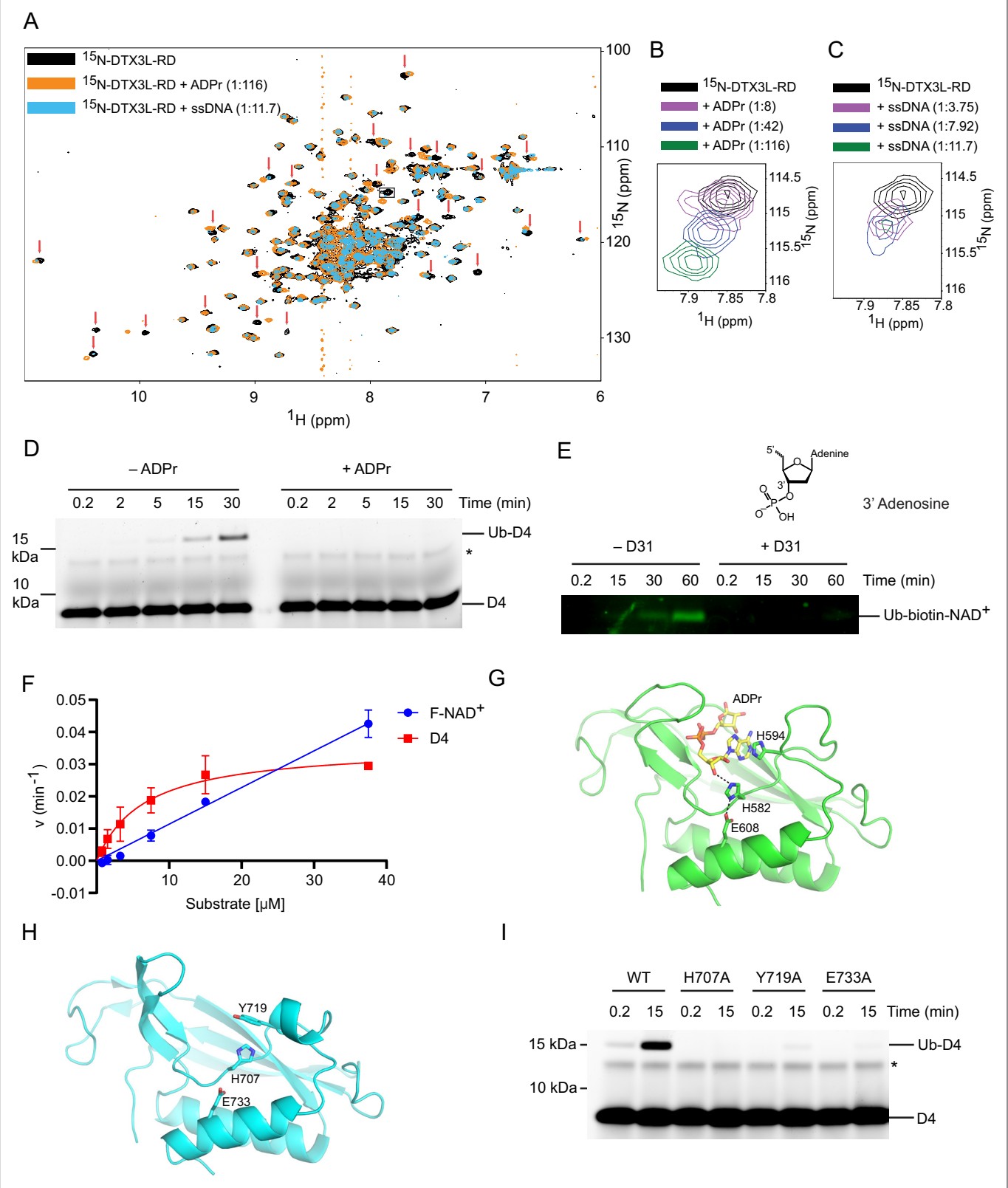

**Figure 4.** DTX3L DTC domain binds and facilitates ubiquitin (Ub)-DNA formation. (**A**) $^1$H-$^{15}$N heteronuclear single-quantum coherence (HSQC) spectra of $^{15}$N-DTX3L-RD (black), ADPr-$^{15}$N-DTX3L-RD (orange), and single-stranded DNA (ssDNA) D30-$^{15}$N-DTX3L-RD (blue). Red arrows indicate cross peaks that shift upon titrating with adenosine 5'-diphosphate (ADP)–ribose (ADPr) or ssDNA. (**B**) Close-up view of the cross peak indicated by the black box in (**A**) upon titration of specified molar ratios of ADPr with $^{15}$N-DTX3L-RD. (**C**) Close-up view of the cross peak indicated by the black arrow in (**A**) upon

*Figure 4 continued on next page*

*Figure 4 continued*

titration of specified molar ratios of ssDNA D30 with $^{15}$N-DTX3L-RD. (**D**) Fluorescently detected SDS-PAGE gel of in vitro ubiquitination of 6-FAM-labelled ssDNA D4 by DTX3L-RD in the presence of E1, UBE2D2, Ub, Mg$^{2+}$-ATP and treated with excess ADPr. (**E**) Western blot of in vitro ubiquitination of biotin-NAD$^+$ by DTX3L-RD in the presence of E1, UBE2D2, Ub, Mg$^{2+}$-ATP and treated with excess ssDNA D31. (**F**) Kinetics of Ub-D4 and Ub-F-NAD$^+$ formation catalysed by DTX3L-RD. Data from two independent experiments (n=2) were fitted with the Michaelis–Menten equation and $k_{cat}/K_m$ value for D4 (5457 M$^{-1}$min$^{-1}$) was calculated. $k_{cat}/K_m$ value for F-NAD$^+$ (1190 M$^{-1}$min$^{-1}$) was estimated from the slope of the linear portion of the curve. (**G**) Structure of DTX2-DTC domain (green) bound to ADPr (yellow) (PDB: 6Y3J). The sidechains of H582, H594, and E608 are shown in sticks. Hydrogen bonds are indicated by dotted lines. (**H**) Structure of DTX3L-DTC domain (cyan; PDB: 3PG6). The sidechains of H707, Y719, and E733 are shown in sticks. (**I**) Fluorescently detected SDS-PAGE gel of in vitro ubiquitination of 6-FAM-labelled ssDNA D4 by full length DTX3L WT, H707A, Y719A, and E733A in the presence of E1, UBE2D2, Ub, Mg$^{2+}$-ATP. Asterisks in (**D**) and (**I**) indicate contaminant band from ssDNA. Raw unedited and uncropped gel images of (**D**), (**E**) and (**I**) are shown in *Figure 4—source data 1 and 2*, respectively. Data points for (**F**) are shown in *Figure 4—source data 3*.

The online version of this article includes the following source data and figure supplement(s) for figure 4:

**Source data 1.** Raw unedited gels for *Figure 4*.

**Source data 2.** Uncropped and labelled gels for *Figure 4*.

**Source data 3.** Kinetic data related to *Figure 4F*.

**Figure supplement 1.** DTX3L-RD binds adenosine 5'-diphosphate (ADP)–ribose (ADPr) and single-stranded nucleic acids (ssNA).

**Figure supplement 1—source data 1.** Raw unedited gels for *Figure 4—figure supplement 1*.

**Figure supplement 1—source data 2.** Uncropped and labelled gels for *Figure 4—figure supplement 1*.

estimated catalytic efficiency ($k_{cat}/K_m$ = 5457 M$^{-1}$ min$^{-1}$ for D4 and estimated $k_{cat}/K_m$ = 1190 M$^{-1}$ min$^{-1}$ for F-NAD$^+$; *Figure 4F*) suggested that DTX3L-RD exhibited 4.5-fold higher catalytic efficiency for D4 than F-NAD$^+$. This difference primarily results from a better $K_m$ value for D4 compared to F-NAD$^+$. Although DTX3L-RD showed weak $K_m$ for F-NAD$^+$, it displays a higher rate for converting F-NAD$^+$ to Ub-F-NAD$^+$ at higher substrate concentrations (*Figure 4F*). Thus, substrate concentration will play a role in determining the preference.

Previously, we demonstrated the importance of DTX2 His[594] located in the DTC pocket for the interaction with ADPr (*Figure 4G*; *Ahmed et al., 2020*). We examined the effect of mutating the corresponding residue (Tyr[719]) in DTX3L (*Figure 4H*) and found that the Y719A mutant was defective in the ubiquitination of ssDNA (*Figure 4I*). Two catalytic residues in the DTC domain of DTX2 (*Zhu et al., 2022*), His[582] and Glu[608], have been previously identified as important for the ubiquitination of ADPr, and are conserved across the DTX family members (*Figure 4G and H* and *Figure 5—figure supplement 1A*). Based on the similarities in mechanism and conservation of the binding site between ADPr and ssDNA, we next investigated if these catalytic residues were involved in the ubiquitination of ssDNA. Substitution of His[707] by alanine in FL DTX3L completely abolished the formation of Ub-DNA, whilst substitution of Glu[733] by alanine considerably impaired product formation (*Figure 4I*). To test whether this mechanism was conserved across ssNAs, we repeated the mutant DTX3L ubiquitination assay with ssRNA R4 and found similar results (*Figure 4—figure supplement 1H*). Overall, the data support the idea that DTX3L utilises the ADPr-binding pocket and the same catalytic residues in the DTC domain to bind and catalyse the ubiquitination of ssNAs.

## Ubiquitination of ssDNA is unique to DTX3 and DTX3L

Since the DTC domains are conserved between DTX family members and share the ability to ubiquitinate ADPr (*Chatrin et al., 2020*), we tested whether the ubiquitination of ssDNA was a universal feature of the DTX family. Unexpectedly, only DTX3L and to a lesser extent DTX3 were able to ubiquitinate ssDNA under our assay conditions (*Figure 5A–D*). Several potential explanations were considered for this preference: (1) other DTX DTCs might not bind ssDNA; (2) other DTXs might favour different sequences of ssDNA; (3) or the arrangement of the RING and DTC domains in relation to each other might play a crucial role in catalysis. To probe this further, we titrated DTX3L-RING with excess DTX3L-DTC and tracked the formation of Ub-DNA, comparing the ability of the separate domains to ubiquitinate DNA in trans with that of DTX3L-RD. No ubiquitination of ssDNA was observed even at a 10:1 molar ratio of DTC to RING domain (*Figure 5E*), indicating that the ubiquitination of ssDNA only occurs at an appreciable rate in cis. Next, we examined whether DTX2 could catalyse the ubiquitination of ssDNA using 9 different ssDNA sequences and found that DTX2 was unable to ubiquitinate them (*Figure 5F*). Lastly, we produced domain-swapped variants of DTX2 and DTX3L in which their

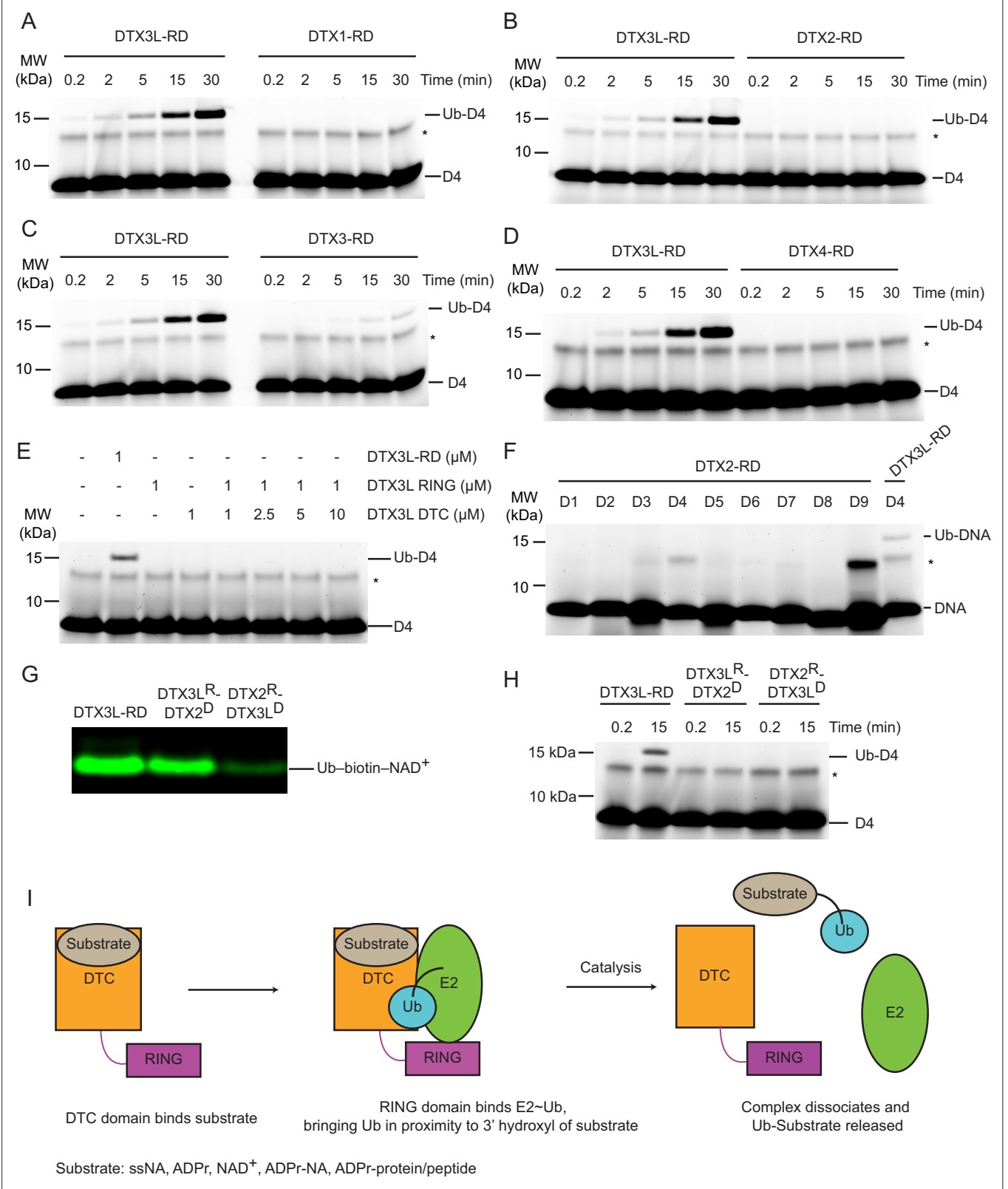

**Figure 5.** Select DTX RING-DTC domains catalyse ubiquitination of ssDNA. (**A**) Fluorescently detected SDS-PAGE gel of in vitro ubiquitination of 6-FAM-labelled ssDNA D4 by DTX3L-RD or DTX1-RD in the presence of E1, UBE2D2, Ub, $Mg^{2+}$-ATP. (**B**) As in (**A**) but with DTX2-RD. (**C**) As in (**A**) but with DTX3-RD. (**D**) As in (**A**) but with DTX4-RD. (**E**) As in (**A**) but with DTX3L-RD or DTX3L RING with increasing concentrations of DTX3L DTC. (**F**) Fluorescently detected SDS-PAGE gel of in vitro ubiquitination of 6-FAM-labelled ssDNA D1-9 by DTX2-RD. A reaction with DTX3L-RD and 6-FAM-

*Figure 5 continued on next page*

*Figure 5 continued*

labelled ssDNA D4 was included as a positive control. (**G**) Western blot of in vitro ubiquitination of biotin-NAD$^+$ in the presence of E1, UBE2D2, Ub, Mg$^{2+}$-ATP, NAD$^+$, biotin-NAD$^+$ with either DTX3L-RD, DTX3L$^R$-DTX2$^D$, or DTX2$^R$-DTX3L$^D$ and separated by SDS-PAGE. (**H**) Fluorescently detected SDS-PAGE gel of in vitro ubiquitination of 6-FAM-labelled ssDNA D4 by DTX3L-RD, DTX3L$^R$-DTX2$^D$, or DTX2$^R$-DTX3L$^D$ in the presence of E1, UBE2D2, Ub, Mg$^{2+}$-ATP. (**I**) Schematic diagrams showing the proposed mechanism of ubiquitination of substrates by DTX3L-RD. Asterisks in (A–F) and (**H**) indicate contaminant bands from ssDNA. Raw unedited and uncropped gel images of (A–H) are shown in *Figure 5—source data 1 and 2*, respectively.

The online version of this article includes the following source data and figure supplement(s) for figure 5:

**Source data 1.** Raw unedited gels for *Figure 5*.

**Source data 2.** Uncropped and labelled gels for *Figure 5*.

**Figure supplement 1.** Properties of DELTEX (DTX) family DELTEX C-terminal (DTC) domains.

respective RING and DTC domains were swapped (DTX2$^R$-DTX3L$^D$ and DTX3L$^R$-DTX2$^D$) and examined the ability of these chimeras to ubiquitinate biotin-NAD$^+$ or ssDNA D4. We hypothesised that the DTX3L DTC domain of DTX2$^R$-DTX3L$^D$ would be sufficient to enable the binding and ubiquitination of ssDNA. Both chimeras were competent in performing ubiquitination of biotin-NAD$^+$ (*Figure 5G*) but were unable to produce Ub-DNA (*Figure 5H*). These results suggest that the specific arrangement of the RING and DTC domains in DTX3L is crucial in specifying Ub modification of ssDNA. Taken together, we propose a mechanism for the ubiquitination of ssNAs by the DTX3L-RD domains (*Figure 5I*): the DTC domain binds ssNAs, the RING domain recruits E2~Ub, and the RD domains bring the thioester linked Ub into proximity with the 3′ end of ssNA. This allows the catalytic residues in the DTC domain to enable the covalent attachment of Ub to the 3′ hydroxyl group on the 3′-end of ssNA. Subsequently, the complex dissociates, releasing ssNA-Ub. The substrates for the DTC domain also include ADPr, NAD$^+$, and ADPr-modified nucleic acids and proteins.

## Discussion

The diversity of non-proteinaceous substrates that have been demonstrated to undergo ubiquitination has been steadily expanding over the last few years (*Otten et al., 2021*; *Chatrin et al., 2020*; *Kelsall et al., 2019*). Here, we report additional examples of non-canonical substrates, namely ssRNA, ssDNA, and dsDNA with a 3′ overhang of at least two nucleotides, that undergo ubiquitination by DTX3L. Our biochemical and NMR analyses show that the DTX3L DTC domain binds ssNAs, whilst the RING domain is required for the formation of the Ub-NA product. Mutational studies herein reveal three residues, His$^{707}$, Tyr$^{719}$, and Glu$^{733}$, within the DTC domain that are required for the formation of Ub-NA. These residues are conserved within the DTC domain, are known to coordinate ADPr (*Chatrin et al., 2020*; *Ahmed et al., 2020*), and are proposed to constitute a catalytic triad essential for DTX E3-mediated ubiquitination of ADPr (*Zhu et al., 2022*). These findings, coupled with the knowledge that the 3′-hydroxyl of the nucleotide ribose is the modification site, suggest a potential similarity in the mechanism of Ub modification of ssDNA, ssRNA, and ADPr by DTX3L.

The chemical structures of ssNAs and ADPr share a ribose-5-phosphate moiety and potentially an adenine base, depending on the ssNA sequence. While our data suggest that DTX3L prefers to ubiquitinate ssDNA ending with adenosine and to some extent guanosine – both purine bases, pyrimidine bases are also tolerated. This raises uncertainty about how DTX3L DTC binds ssNA. Our experiments with dsDNA with a 3′ overhang in one strand showed that dsDNA with a single nucleotide overhang cannot be ubiquitinated, whereas dsDNA with a two-nucleotide overhang was ubiquitinated. Notably, alterations in the sequence of the last two nucleotides at the 3′ end influenced Ub-DNA product formation (*Figure 3A and B*). These observations suggest that the DTX3L DTC NA-binding pocket may accommodate a dinucleotide. Upon examining the ssDNA sequences utilised in our study (*Figure 1F*; *Figure 3A*; *Table 1*), it appears that the DTX3L DTC pocket might accommodate diverse sequences in the last two nucleotides at the 3′ end, with adenine being the preferred last nucleotide and cytosine being the least favoured. Despite the sequence conservation of the RING-DTC domain with other DTX proteins, our biochemical data revealed that only DTX3L and DTX3 were able to ubiquitinate ssDNA. A comparison of the aligned sequences of the DTX RING-DTC domains reveals some differences (*Figure 5—figure supplement 1A*). The RING domains of DTX3L and DTX3 lack insertions found in DTX1, DTX2, and DTX4, making their RING domains appear smaller (*Figure 5—figure supplement 1A–C*). Furthermore, DTX3L and DTX3 lack an AR motif found near the DTC pocket. The

absence of the AR motif causes a slight conformational change near the DTC pocket, resulting in an extended β-strand and the loss of a bulge containing the arginine residue (*Figure 5—figure supplement 1A,D, E*). Consequently, DTX3L has an extended groove adjacent to the DTC pocket that could accommodate one or more of the nucleotides 5' to the targeted terminal nucleotide. Replacing DTX2-RD's DTC domain with that of DTX3L produced a chimeric construct that was able to ubiquitinate NAD$^+$ but not ssDNA (*Figure 5G and H*), suggesting that additional factors beyond the catalytic residues and ssDNA binding properties of the DTC domain likely play a role in ubiquitination. Our findings show that this reaction only occurs in cis, pointing to the orientation of the RING relative to the DTC domain as an important feature in the formation of Ub-DNA. Future structural characterisation of DTX3L-RD binding to ssDNA or ssRNA should provide insights into its binding mode and sequence selectivity, thereby potentially revealing whether the biological substrate is ssDNA, ssRNA, or both.

Typically, KH domains contain a GXXG motif within the loop between the first and second α helix (*Grishin, 2001*). However, analysis of the sequence of the KHL domains in DTX3L shows these domains lack this motif. Multiple studies have shown that mutation in this motif abolishes binding to nucleic acids (*Hollingworth et al., 2012*; *Talwar et al., 2017*; *Yadav et al., 2021*; *Lin et al., 2012*). Our findings show the DTX3L DTC domain binds nucleic acids but whether the KHL domains contribute to nucleic acid binding requires further investigation. Additionally, the structure of the first KHL domain was recently reported and shown to form a tetrameric assembly (*Vela-Rodríguez et al., 2024*). Our analysis with DTX3L 232-C, which lacks the first KHL domain and RRM, indicates that it can still bind ssDNA and ssRNA. Despite this, a more detailed analysis will be required to determine whether oligomerisation plays a role in nucleic acid binding and ubiquitination.

In recent years, several studies have unveiled diverse reactions catalysed by DTX3L. It forms a heterodimeric complex with PARP9, facilitating the direct ubiquitination of substrates such as H2B (*Zhang et al., 2015*). This complex can recruit PARylated substrates, either through PARP9 macrodomains (*Zhang et al., 2015*) or DTX3L DTC domain (*Ahmed et al., 2020*), for direct substrate ubiquitination. Notably, it can catalyse the transfer of Ub to NAD$^+$ or ADPr (*Chatrin et al., 2020*; *Yang et al., 2017*; *Zhu et al., 2022*), including ADPr moieties of PARylated substrates, encompassing both proteins and nucleic acids (*Zhu et al., 2022*). Moreover, we have expanded its substrate repertoire by demonstrating its ability to directly ubiquitinate the 3' end of ssNAs. Whether this reaction occurs in cells and what the native substrate (RNA, DNA, or both) are yet to be determined. Due to the labile nature of the ribose ester bond and cleavage of Ub conjugates by DUBs, as well as the digestion of nucleic acids by nucleases, our ability to probe the Ub modification of ssNAs in a cellular context has so far been limited. Connecting these chemical reactions to precise biological functions requires carefully designed experiments aimed at distinguishing these functions. Based on the known functions of the DTX3L/PARP9 complex and the findings of this study, we propose several hypotheses for future research. The DTX3L/PARP9 complex has a known involvement in DNA damage repair, utilising PARP9's macrodomains to recognise PAR and assist in complex recruitment to the damaged foci (*Yang et al., 2017*). RNA itself contributes to DNA damage repair (*Bader et al., 2020*). Likewise, these damage sites may involve DNA breaks with 3' overhangs. Our findings reveal that DTX3L can recognise and ubiquitinate ssRNA, or dsDNA with a 3' overhang of two or more nucleotides. This observation suggests a possible substrate for DTX3L in DNA damage repair pathways, which requires further investigation. In eukaryotic cells, messenger RNA (mRNA) is a source of ssRNA that commonly possesses a poly-A tail at the 3' end. Our data indicated that DTX3L can ubiquitinate ssRNA containing a poly-A sequence (*Figure 1G*, lane R9), offering a possibility for future exploration. Viruses contain a diverse range of genetic materials including ssNAs, which might be plausible ubiquitination targets. Studies have shown that expression of DTX3L/PARP9 complex is induced after viral infection, and the complex plays a role in antiviral defense mechanisms by ubiquitinating both host and viral proteins (*Zhang et al., 2015*; *Huang et al., 2023*). Ubiquitination of viral ssNAs by DTX3L presents another intriguing hypothesis for future research.

Our study reports the first example of direct ubiquitination of ssNAs by DTX3L and illustrates the possibility of additional roles for ubiquitination beyond that of a conventional post-translational modification.

# Materials and methods

## Key resources table

| Reagent type (species) or resource | Designation | Source or reference | Identifiers | Additional information |
|---|---|---|---|---|
| Recombinant DNA reagent (*Homo sapiens*) | pGEX4T-3 TEV DTX3L FL | This paper | Uniprot: Q8TDB6-1 | Codon-optimised synthetic gene |
| Recombinant DNA reagent (*Homo sapiens*) | pGEX4T-3 TEV DTX3L FL mutants | This paper | | Codon-optimised synthetic gene H707A Y719A E733A |
| Recombinant DNA reagent (*Homo sapiens*) | pGEX4T-3 TEV DTX3L 232-C | This paper | | Codon-optimised synthetic gene |
| Recombinant DNA reagent (*Homo sapiens*) | pGEX4T-3 TEV DTX3L 607-C | This paper | | Codon-optimised synthetic gene |
| Recombinant DNA reagent (*Homo sapiens*) | pGEX4T-3 TEV DTX3L 544–606 | This paper | | Codon-optimised synthetic gene |
| Recombinant DNA reagent (*Homo sapiens*) | DTX1 388-C | *Chatrin et al., 2020* | | |
| Recombinant DNA reagent (*Homo sapiens*) | DTX2 390-C | *Chatrin et al., 2020* | | |
| Recombinant DNA reagent (*Homo sapiens*) | DTX3 148-C | *Chatrin et al., 2020* | | |
| Recombinant DNA reagent (*Homo sapiens*) | DTX4 387-C | *Chatrin et al., 2020* | | |
| Recombinant DNA reagent (*Homo sapiens*) | DTX3L 544-C | *Chatrin et al., 2020* | | |
| Recombinant DNA reagent (*Homo sapiens*) | UBA1 | *Nakasone et al., 2022* | | |
| Recombinant DNA reagent (*Homo sapiens*) | UBE2D2 | *Dou et al., 2012* | | |
| Recombinant DNA reagent (*Homo sapiens*) | Ub | *Gabrielsen et al., 2017*; *Volk et al., 2005* | | |
| Recombinant DNA reagent (*Homo sapiens*) | Fluorescent Ub | *Magnussen et al., 2020* | | |
| Recombinant DNA reagent (*Homo sapiens*) | DTX3L(232-C)/PARP9(509-C) | This paper | Uniprot PARP9: Q8IXQ6-1 | Codon-optimised synthetic gene |
| Recombinant DNA reagent (*Homo sapiens*) | pRSFDuet 12 X His TEV PARG 448-C | *Tucker et al., 2012* | | Codon-optimised synthetic gene |
| Recombinant DNA reagent (*Homo sapiens*) | USP2 260-C | *Chatrin et al., 2020* | | |
| Strain, strain background (*Escherichia coli*) | BL21(DE3) Gold | Agilent | Cat# 230132 | Chemically competent |

*Continued on next page*

*Continued*

| Reagent type (species) or resource | Designation | Source or reference | Identifiers | Additional information |
|---|---|---|---|---|
| Strain, strain background (*Escherichia coli*) | Rosetta 2(DE3)pLysS | Merck Millipore Novagen | Cat# 71403 | Chemically competent |
| Peptide, recombinant protein | Neutravidin Protein, DyLight 800 | Invitrogen | Cat# 22853 | WB (1:10000) |
| Sequence-based reagent | Assorted oligos | Integrated DNA Technologies | | Refer to *Table 1* for sequences |
| Peptide, recombinant protein | Benzonase Nuclease | Merck | Cat# E1014 | |
| Chemical compound, drug | $NH_2OH$ | Merck Millipore | Cat# 467804 | |
| Chemical compound, drug | ADPr | Merck | Cat# A0752 | |
| Chemical compound, drug | Biotin -$NAD^+$ | Biolog | Cat# N 012 | |
| Chemical compound, drug | F-$NAD^+$ | Biolog | Cat# N 023 | |
| Software, algorithm | GraphPad Prism | | RRID:SCR_002798 | https://www.graphpad.com/features |
| Software, algorithm | Quantity One 1-D analysis software | Bio-Rad | RRID:SCR_014280 | https://www.bio-rad.com/en-uk/product/quantity-one-1-d-analysis-software?ID=1de9eb3a-1eb5-4edb-82d2-68b91bf360fb |

## Construct generation

Constructs were generated by standard polymerase chain reaction–ligation techniques and verified by automated sequencing. GST-tagged constructs were cloned into a modified form of pGEX4T-3 (GE Healthcare) with a TEV cleavage site and a second ribosomal binding and multiple cloning site (pABLO TEV) and His-tagged constructs were cloned into a modified form of pRSF_Duet-1 (Novagen) with a TEV cleavage site following the hexahistidine tag. All proteins are from human sequences. RD domains are comprised of residues 388-C of DTX1, residues 390-C of DTX2, residues 148-C of DTX3, residues 387-C of DTX4, and residues 544-C of DTX3L. DTC of DTX3L comprises residues 607-C. PARG comprises residues 448-C. USP2 comprises residues 260-C. The complex, DTX3L(232-C)/PARP9(509-C), was cloned into a bicistronic vector in which PARP9(509-C) was cloned into the first MCS of pRSF_Duet-1 with an N-terminal His-MBP tag followed by a TEV cleavage site, and DTX3L(232-C) was cloned into the second MCS untagged. DTX3L 232-C and PARP9 509-C were sufficient to form complex (*Zhang et al., 2015*). DTX3L FL and DTX3L 232-C were cloned into the pABLO vector.

## Protein expression and purification

Expression of recombinant proteins was conducted in *Escherichia coli* BL21(DE3) Gold or Rosetta2(DE3) pLysS cells. Cultures were grown in LB broth (Miller) at 37 °C until an $OD_{600}$ of 0.6–0.8 was reached at which point expression was induced with 0.2 mM isopropyl β-D-1-thiogalactopyranoside (IPTG) at 20 °C for 12–16 hr. $^{15}N$-labelled DTX3L-RD was obtained using M9 minimal medium. Briefly, 20 mL starter cultures were grown in LB medium overnight before cells were pelleted and washed in M9 medium. Each pellet was added to 1 L of M9 medium supplemented with: 1 g of $^{15}NH_4Cl$, 5 g glucose, 50 mg kanamycin, 1 X Vitamin Stock (Gibco), 1 mg D-Biotin and trace metals. Cells were grown until an $OD_{600}$ of 0.6–0.8 was reached at which point expression was induced with IPTG at 20 °C for 20 hr.

Cells were harvested by centrifugation following expression and lysed by microfluidizer or sonicator. Cells expressing GST-tagged proteins were resuspended in 50 mM Tris-HCl (pH 7.6), 200 mM NaCl, and 1 mM DTT. Cells expressing His-tagged proteins were resuspended in 25 mM Tris-HCl (pH 7.6), 200 mM NaCl, 15 mM imidazole, 5 mM beta-mercaptoethanol. Cell lysates were cleared by high-speed ultracentrifugation. Clarified lysates were applied to glutathione affinity or $Ni^{2+}$-agarose

by incubating for 1–2 hr on a rotary shaker at 4 °C or using a gravity column. Beads were washed in buffers similar to the lysis buffer. $^{15}$N-DTX3L-RD was eluted with 6xHis intact in 25 mM Tris-HCl (pH 7.6), 200 mM NaCl, 5 mM BME, and 200 mM imidazole, then further purified by ion exchange chromatography, followed by size exclusion chromatography on a Superdex 75 column (GE Healthcare) into buffer containing 20 mM sodium phosphate, (pH 7.0), 100 mM NaCl, 0.02% NaN$_3$ and 1 mM TCEP before snap-freezing in liquid nitrogen and storing at −80 °C.

For removal of the GST tag, protein samples were dialysed against 25 mM Tris-HCl (pH 7.6), 150 mM NaCl, and 5 mM BME overnight at 4 °C in the presence of TEV protease or alternatively cleaved on the beads with TEV from a glutathione agarose column in 50 mM Tris-HCl (pH 8.0), 200 mM NaCl, and 5 mM DTT. The dialysed and cleaved samples were loaded onto the same resin and flow through collected to the separate from the affinity tags or the remaining uncleaved proteins. Additional purification was performed by size exclusion chromatography on a Superdex 75 column or Superdex 200 column (GE Healthcare), depending on protein size, into 25 mM Tris-HCl (pH 7.6), 150 mM NaCl, and 1 mM DTT before snap-freezing in liquid nitrogen and storing at −80 °C. Protein concentrations were determined using Bio-RAD protein assay.

DTX3L (232-C)/PARP9 (509-C) complex was obtained by expressing the bicistronic construct of His-MBP-PARP9(509-C)/DTX3L(232C). The complex was purified by Ni$^{2+}$-agarose, cleaved by TEV treatment, loaded back on Ni$^{2+}$ for tag and protease removal, and further purified by Source S cation exchange chromatography and Superdex 200.

Additional proteins: *Homo sapiens* Uba1 (*Nakasone et al., 2022*), UBE2D2 (*Dou et al., 2012*), Ub (*Gabrielsen et al., 2017*; *Volk et al., 2005*), PARG (*Tucker et al., 2012*), and USP2 (*Renatus et al., 2006*) were purified as in previously described protocols.

## Oligonucleotides

Oligonucleotides were obtained from Integrated DNA Technologies and are listed in *Table 1*.

## Fluorescence polarisation assay

50 nM 6-FAM-labelled nucleic acids (Integrated DNA Technologies) were incubated with 0–5 µM proteins in reaction buffer (20 mM Hepes pH 7.0, 1 mM MgCl$_2$, 50 mM NaCl, filter sterilised). Fluorescence polarisation measurements were taken using Tecan Spark ($\lambda$ ex = 485 nm, $\lambda$ em = 535 nm). Fold change was calculated based on the change in polarisation compared to the fluorescent ligand in the absence of protein. Data were plotted in Prism (GraphPad).

## DNA and RNA ubiquitination assays

UBA1 (0.2 µM), UBE2D2 (5 µM), and Ub (50 µM) were incubated in 50 mM Tris-HCl, 5 mM MgCl$_2$, 50 mM NaCl, and 5 mM ATP at room temperature for 15 min to allow for generation of E2~Ub. DTX (1 µM) and nucleic acid (3 µM) were incubated together at room temperature for 10 min prior to mixing with other components. After the components were combined, reactions were incubated at 37 °C. Aliquots were taken immediately (indicated by 0.2 min time point) and at specified subsequent time points. The reactions were halted by the addition of 2 X Loading Dye containing 250 mM DTT and resolved by SDS-PAGE (NuPAGE 4–12% Bis-Tris, Invitrogen). For FAM-labelled nucleic acids, gels were visualised using Typhoon FLA 700 with $\lambda$ ex = 473 nm and Y520 emission filter (Cytiva Life Sciences). For IRDye 800 labelled nucleic acids, gels were scanned with an Odyssey CLx Imaging System (LI-COR Biosciences). For *Figures 1J and 5A–C* and D, 25 µM D4 was used. For *Figure 1L*, 25 µM D11 was used. For assays with USP2, Benzonase and PARG, reactions were conducted as above and after UBA1 (0.2 µM), UBE2D2 (5 µM), Ub (50 µM), DTX3L-RD (2 µM), and D4 or R4 (2 µM) were incubated together for 30 min, reactions were quenched with 250 mM DTT and buffer exchanged into 25 mM Tris-HCl (pH 7.6), 150 mM NaCl, 5 mM MgCl$_2$, and 1 mM DTT using a 7 kDa Zeba Spin column. USP2 (1.35 µM), Benzonase (1 µM), or PARG (1 µM) was added and incubated for a further 30 min at 37 °C. Subsequently, an aliquot was mixed with 2 X Loading Dye and resolved by SDS-PAGE. For assays conducted at pH 9.5, reactions were conducted as above and after UBA1 (0.2 µM), UBE2D2 (5 µM), Ub (50 µM), DTX3L-RD (1 µM), and D4 (3 µM) were incubated together for 30 min, reactions were quenched with 250 mM DTT and buffer exchanged into 25 mM Tris-HCl (pH 9.5), 150 mM NaCl and 1 mM DTT using a 7 kDa Zeba Spin column. Then 2 mM EDTA was added, and the reaction was incubated at 37 °C. Aliquots were taken at indicated time points, mixed with 2 X Loading Dye, and

resolved by SDS-PAGE. For assays conducted with $NH_2OH$ treatment, reactions were conducted as above. After UBA1 (0.2 µM), UBE2D2 (5 µM), Ub (50 µM), DTX3L-RD (1 µM), and D4 (3 µM) were incubated together for 30 min, reactions were quenched with 250 mM DTT and buffer was exchanged into 25 mM Tris-HCl (pH 9), 150 mM NaCl, and 1 mM DTT using a 7 kDa Zeba Spin column. Then 1.5 M $NH_2OH$ was added, and the reaction was incubated at 37 °C. Aliquots were taken at indicated time points, mixed with 2 X Loading Dye, and resolved by SDS-PAGE.

### Autoubiquitination assay

UBA1 (0.2 µM), UBE2D2 (5 µM), and fluorescently-labelled Ub (*Magnussen et al., 2020*) (25 µM) were incubated in 50 mM Tris-HCl, 5 mM $MgCl_2$, 50 mM NaCl, and 5 mM ATP at room temperature for 15 min to allow for the generation of E2~Ub. DTX3L (1 µM) was added and reactions were incubated at 20 °C. Aliquots were taken at indicated time points, and reactions were halted by the addition of 2 X Loading Dye and resolved by SDS-PAGE. Gels were scanned with an Odyssey CLx Imaging System.

### Lysine discharge assay

UBA1 (0.5 µM), UBE2D2 (25 µM), and Ub (25 µM) were incubated in 50 mM Tris-HCl, 5 mM $MgCl_2$, 50 mM NaCl, and 5 mM ATP at room temperature for 15 min to allow for generation of E2~Ub. Charged reactions were stopped with 30 mM EDTA and T0 samples taken. DTX3L (1 µM) and lysine (300 mM) were added, and reactions were incubated at room temperature. Aliquots were taken at indicated time points, and reactions were halted by the addition of 2 X Loading Dye and resolved by SDS-PAGE. Gels were scanned with a ChemiDoc (Bio-Rad).

### Competition ubiquitination assays

UBA1 (0.2 µM), UBE2D2 (5 µM), and Ub (50 µM) were incubated in 50 mM Tris-HCl, 5 mM $MgCl_2$, 50 mM NaCl, and 5 mM ATP at room temperature for 15 min to allow for generation of E2~Ub. DTX3L-RD (1 µM) and D4 (3 µM) were incubated together at room temperature for 10 min prior to mixing with other components. After the components were combined, 1 mM ADPr was added and reactions were incubated at 37 °C. Aliquots were taken at indicated time points, reactions were halted by the addition of 2 X Loading Dye containing 250 mM DTT and resolved by SDS-PAGE. Gels were visualised using Typhoon FLA 700 with $\lambda$ ex = 473 nm and a Y520 emission filter. For *Figure 4E*, DTX3L-RD (1 µM) and biotin-$NAD^+$ (5 µM) were incubated together at room temperature for 10 min prior to mixing with other components. After components were combined, 2 mM D31 was added and reactions were incubated at 30 °C. Aliquots were taken at indicated time points, reactions were halted by the addition of 2 X Loading Dye containing 250 mM DTT and resolved by SDS-PAGE. The samples were transferred to a nitrocellulose membrane and blocked in 5% BSA. Membranes were incubated with DyLight 800 conjugated NeutrAvidin (Thermo Fisher Scientific, cat. no. 22853; 1:10,000) and visualised using an Odyssey CLx Imaging System.

### Kinetic analysis of Ub-D4 and Ub-F-NAD⁺ formation

UBA1 (0.2 µM), UBE2D2 (5 µM), and Ub (50 µM) were incubated in 50 mM Tris-HCl, 5 mM $MgCl_2$, 50 mM NaCl, and 5 mM ATP at room temperature for 15 min to allow for generation of E2~Ub. DTX (1 µM) and D4 or F-$NAD^+$ (Biolog Life Science Institute, cat. no. N 023) at varying concentrations (0–37.5 µM) were incubated together at room temperature for 10 min prior to mixing with other components. Aliquots were taken after 15 min, reactions were halted by the addition of 2 X Loading Dye containing 250 mM DTT and resolved by SDS-PAGE. Gels were visualised using Typhoon FLA 700 with $\lambda$ ex = 473 nm and a Y520 emission filter. A known amount of D4 was visualised on the gel alongside D4 ubiquitination reactions. A known amount of F-$NAD^+$ was pipetted onto Whatman filter paper and scanned alongside the gel of F-$NAD^+$ ubiquitination reactions. Ub-D4 and Ub-F-$NAD^+$ were quantified and initial rates were expressed as nmoles of product formed per minute per mole of DTX3L-RD. Data from two independent experiments were fitted to the Michaelis-Menten equation using Prism. The Michaelis-Menten curve for Ub-F-$NAD^+$ was not saturated, therefore, $k_{cat}/K_m$ was estimated from the slope of the linear portion of the curve.

### Biotin-NAD⁺ ubiquitination assay

UBA1 (0.2 µM), UBE2D2 (5 µM), and Ub (50 µM) were incubated in 50 mM HEPES-NaOH, pH 7.5, 5 mM $MgCl_2$, 50 mM NaCl, and 5 mM ATP at room temperature for 10 min to allow for generation of

E2~Ub. DTX (1 µM), NAD$^+$ (200 µM) and biotin-NAD$^+$ (10 µM) were added and reactions were incubated at 20 °C for 1 hr. Reactions were halted by the addition of 3 X Loading Dye containing 250 mM DTT and resolved by SDS-PAGE. The samples were transferred to a nitrocellulose membrane and blocked in 5% BSA. Membranes were incubated with DyLight 800 conjugated NeutrAvidin (Thermo Fisher Scientific, cat. no. 22853; 1:10,000) and visualised using an Odyssey CLx Imaging System.

## Solution NMR experiments

All NMR data were acquired using a Bruker Avance III 600-MHz spectrometer equipped with a cryogenic triple resonance inverse probe. DTX3L-RD samples were prepared at ~100 µM in 20 mM sodium phosphate, (pH 7.0), 100 mM NaCl, 0.02% NaN$_3$, 1 mM TCEP, and 5% D$_2$O with 0.00025% TSP. Experiments were carried out at 298 K, and $^1$H-$^{15}$N HSQC spectra were recorded with 16 scans using 200 complex points with a sweep width of 36 parts per million (ppm) in the $^{15}$N dimension. All spectra were processed with 256 points in the indirect dimension using Bruker TopSpin 3.5 and analysed using CcpNmr AnalysisAssign (*Skinner et al., 2016*).

## Acknowledgements

We thank Core Services and Advanced Technologies at the Cancer Research UK Scotland Institute (C596/A17196 and A31287) with particular thanks to Molecular Technology services; Catherine Winchester for her assistance in critically reviewing this manuscript. This work was supported by Cancer research UK core grants to DTH (A29256) and MB (A29252).

## Additional information

### Competing interests

Danny T Huang: D.T.H is a consultant for Triana Biomedicines. The other authors declare that no competing interests exist.

### Funding

| Funder | Grant reference number | Author |
| --- | --- | --- |
| Cancer Research UK | A29256 | Danny T Huang |
| Cancer Research UK | A29252 | Martin Bushell |
| Cancer Research UK Scotland Institute | C596/A17196 | Danny T Huang |
| Cancer Research UK Scotland Institute | A31287 | Danny T Huang |

The funders had no role in study design, data collection and interpretation, or the decision to submit the work for publication.

### Author contributions

Emily L Dearlove, Conceptualization, Data curation, Formal analysis, Validation, Investigation, Methodology, Writing – original draft, Writing – review and editing; Chatrin Chatrin, Conceptualization, Data curation, Formal analysis, Validation, Investigation, Methodology, Writing – review and editing; Lori Buetow, Data curation, Formal analysis, Validation, Investigation, Methodology, Writing – review and editing; Syed F Ahmed, Formal analysis, Investigation, Methodology, Writing – review and editing; Tobias Schmidt, Resources, Formal analysis, Methodology, Writing – review and editing; Martin Bushell, Resources, Supervision, Funding acquisition, Writing – review and editing; Brian O Smith, Formal analysis, Supervision, Investigation, Writing – review and editing; Danny T Huang, Conceptualization, Data curation, Formal analysis, Supervision, Funding acquisition, Writing – original draft, Project administration, Writing – review and editing

### Author ORCIDs

Syed F Ahmed ⓘ https://orcid.org/0000-0003-1033-2538

Danny T Huang ⓘ https://orcid.org/0000-0002-6192-259X

Reviewer #1 (Public Review): https://doi.org/10.7554/eLife.98070.3.sa1
Reviewer #2 (Public Review): https://doi.org/10.7554/eLife.98070.3.sa2
Author response https://doi.org/10.7554/eLife.98070.3.sa3

## Additional files

### Supplementary files
• MDAR checklist

### Data availability
Original and uncropped images are presented in source data figures. Raw NMR data has been deposited into Dryad (https://doi.org/10.5061/dryad.rn8pk0pmv).

The following dataset was generated:

| Author(s) | Year | Dataset title | Dataset URL | Database and Identifier |
| --- | --- | --- | --- | --- |
| Vira K, Yan W | 2024 | DTX3L ubiquitin ligase ubiquitinates single-stranded nucleic acids | https://doi.org/10.5061/dryad.rn8pk0pmv | Dryad Digital Repository, 10.5061/dryad.rn8pk0pmv |

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
