## [Editor Report · eLife assessment]

This **important** study reports the discovery of a novel nucleotide ubiquitylation activity by the DTX3L E3 ligase. **Solid** evidence is presented for ubiquitin attachment to single-stranded oligonucleotides. This very interesting biochemical finding can be used as a starting point for studies to establish relevance in a physiological setting.

---

## [Referee Report · Reviewer #1 (Public Review)]

In the article by Dearlove et al., the authors present evidence in strong support of nucleotide ubiquitylation by DTX3L, suggesting it is a promiscuous E3 ligase with capacity to ubiquitylate ADP ribose and nucleotides. The authors include data to identify the likely site of attachment and the requirements for nucleotide modification.

While this discovery potentially reveals a whole new mechanism by which nucleotide function can be regulated in cells, there are some weaknesses that should be considered. Is there any evidence of nucleotide ubiquitylation occurring cells? It seems possible, but evidence in support of this would strengthen the manuscript. The NMR data could also be strengthened as the binding interface is not reported or mapped onto the structure/model, this seems of considerable interest given that highly related proteins do have the same activity.

The paper is for the most part well well-written and is potentially highly significant

Comments on revised version:

The revised manuscript has addressed many of the concerns raised and clarified a number of points. As a result the manuscript is improved.

The primary concern that remains is the absence of biological function for Ub-ssDNA/RNA and the inability to detect it in cells. Despite this the manuscript will be of interest to those in the ubiquitin field and will likely provoke further studies and the development of tools to better assess the cellular relevance. As a result this manuscript is important.

Minor issue:

Figure 1A - the authors have now included the constructs used but it would be more informative if the authors lined up the various constructs under the relevant domains in the full-length protein.

---

## [Referee Report · Reviewer #2 (Public Review)]

The manuscript by Dearlove et al. entitled "DTX3L ubiquitin ligase ubiquitinates single-stranded nucleic acids" reports a novel activity of a DELTEX E3 ligase family member, DTX3L, which can conjugate ubiquitin to the 3' hydroxyl of single-stranded oligonucleotides via an ester linkage. The findings that unmodified oligonucleotides can act as substrates for direct ubiquitylation and the identification of DTX3 as the enzyme capable of performing such oligonucleotide modification are novel, intriguing, and impactful because they represent a significant expansion of our view of the ubiquitin biology. The authors perform a detailed and diligent biochemical characterization of this novel activity, and key claims made in the article are well supported by experimental data. However, the studies leave room for some healthy skepticism about the physiological significance of the unique activity of DTX3 and DTX3L described by the authors because DTX3/DTX3L can also robustly attach ubiquitin to the ADP ribose moiety of NAD or ADP-ribosylated substrates. The study could be strengthened by a more direct and quantitative comparison between ubiquitylation of unmodified oligonucleotides by DTX3/DTX3L with the ubiquitylation of ADP-ribose, the activity that DTX3 and DTX3L share with the other members of the DELTEX family.

Comment on revised version:

In my opinion, reviewers' comments are constructively addressed by the authors in the revised manuscript, which further strengthens the revised submission and makes it an important contribution to the field. Specifically, the authors perform a direct quantitative comparison of two distinct ubiquitylation substrates, unmodified oligonucleotides and fluorescently labeled NADH and report that kcat/Km is 5-fold higher for unmodified oligos compared to NADH. This observation suggests that ubiquitylation of unmodified oligos is not a minor artifactual side reaction in vitro and that unmodified oligonucleotides may very well turn out to be the physiological substrates of the enzyme. However, the true identity of the physiological substrates and the functionally relevant modification site(s) remain to be established in further studies.

---

## [Author Response]

The following is the authors’ response to the current reviews.

**Public Reviews:**

**Reviewer #1 (Public Review):**
In the article by Dearlove et al., the authors present evidence in strong support of nucleotide ubiquitylation by DTX3L, suggesting it is a promiscuous E3 ligase with capacity to ubiquitylate ADP ribose and nucleotides. The authors include data to identify the likely site of attachment and the requirements for nucleotide modification.While this discovery potentially reveals a whole new mechanism by which nucleotide function can be regulated in cells, there are some weaknesses that should be considered. Is there any evidence of nucleotide ubiquitylation occurring cells? It seems possible, but evidence in support of this would strengthen the manuscript. The NMR data could also be strengthened as the binding interface is not reported or mapped onto the structure/model, this seems of considerable interest given that highly related proteins do have the same activity.The paper is for the most part well well-written and is potentially highly significantComments on revised version:The revised manuscript has addressed many of the concerns raised and clarified a number of points. As a result the manuscript is improved.The primary concern that remains is the absence of biological function for Ub-ssDNA/RNA and the inability to detect it in cells. Despite this the manuscript will be of interest to those in the ubiquitin field and will likely provoke further studies and the development of tools to better assess the cellular relevance. As a result this manuscript is important.

We agree with the reviewer’s assessment.

Minor issue:Figure 1A - the authors have now included the constructs used but it would be more informative if the authors lined up the various constructs under the relevant domains in the full-length protein.

Figure 1 will be fixed in the Version of Record.

**Reviewer #2 (Public Review):**
The manuscript by Dearlove et al. entitled "DTX3L ubiquitin ligase ubiquitinates single-stranded nucleic acids" reports a novel activity of a DELTEX E3 ligase family member, DTX3L, which can conjugate ubiquitin to the 3' hydroxyl of single-stranded oligonucleotides via an ester linkage. The findings that unmodified oligonucleotides can act as substrates for direct ubiquitylation and the identification of DTX3 as the enzyme capable of performing such oligonucleotide modification are novel, intriguing, and impactful because they represent a significant expansion of our view of the ubiquitin biology. The authors perform a detailed and diligent biochemical characterization of this novel activity, and key claims made in the article are well supported by experimental data. However, the studies leave room for some healthy skepticism about the physiological significance of the unique activity of DTX3 and DTX3L described by the authors because DTX3/DTX3L can also robustly attach ubiquitin to the ADP ribose moiety of NAD or ADP-ribosylated substrates. The study could be strengthened by a more direct and quantitative comparison between ubiquitylation of unmodified oligonucleotides by DTX3/DTX3L with the ubiquitylation of ADP-ribose, the activity that DTX3 and DTX3L share with the other members of the DELTEX family.Comment on revised version:In my opinion, reviewers' comments are constructively addressed by the authors in the revised manuscript, which further strengthens the revised submission and makes it an important contribution to the field. Specifically, the authors perform a direct quantitative comparison of two distinct ubiquitylation substrates, unmodified oligonucleotides and fluorescently labeled NADH and report that kcat/Km is 5-fold higher for unmodified oligos compared to NADH. This observation suggests that ubiquitylation of unmodified oligos is not a minor artifactual side reaction in vitro and that unmodified oligonucleotides may very well turn out to be the physiological substrates of the enzyme. However, the true identity of the physiological substrates and the functionally relevant modification site(s) remain to be established in further studies.

We agree with the reviewer’s assessment.

The following is the authors’ response to the original reviews.

**Public Reviews:**

**Reviewer #1 (Public Review):**
In the article by Dearlove et al., the authors present evidence in strong support of nucleotide ubiquitylation by DTX3L, suggesting it is a promiscuous E3 ligase with capacity to ubiquitylate ADP ribose and nucleotides. The authors include data to identify the likely site of attachment and the requirements for nucleotide modification.While this discovery potentially reveals a whole new mechanism by which nucleotide function can be regulated in cells, there are some weaknesses that should be considered. Is there any evidence of nucleotide ubiquitylation occurring cells? It seems possible, but evidence in support of this would strengthen the manuscript. The NMR data could also be strengthened as the binding interface is not reported or mapped onto the structure/model, this seems of considerable interest given that highly related proteins do have the same activity.The paper is for the most part well well-written and is potentially highly significant, but it could be strengthened as follows:(1) The authors start out by showing DTX3L binding to nucleotides and ubiquitylation of ssRNA/DNA. While ubiquitylation is subsequently dissected and ascribed to the RD domains, the binding data is not followed up. Does the RD protein alone bind to the nucleotides? Further analysis of nucleotide binding is also relevant to the Discussion where the role of the KH domains is considered, but the binding properties of these alone have not been analysed.

We thank the reviewer for the suggestion. We have tested DTX3L RD for ssDNA binding using NMR (see Figure 4A and Figure S2), which showed that DTX3L RD binds ssDNA. We have now tested the DTX3L KH domains for RNA/ssDNA binding using an FP experiment. However, the FP experiment did not show significant changes upon titrating RNA/ssDNA, suggesting that the KH domains alone are not sufficient to bind RNA/ssDNA. We have opted to put this data in the response-to-review as future investigation will be required to examine whether other regions of DTX3L cooperate with RD to bind RNA/ssDNA. We have revised the Discussion on the KH domains. We now state that “Our findings show the DTX3L DTC domain binds nucleic acids but whether the KHL domains contribute to nucleic acid binding requires further investigation.”

**Author response image 1. sa3fig1:** Fold change of fluorescence polarisation of 6-FAM-labelled ssDNA D4 upon titrating with DTX3L variants. DTX3L KH domain fragments were expressed with a N-terminal His-MBP tag to increase the molecular weight to enhance the signal.

(2) With regard to the E3 ligase activity, can the authors account for the apparent decreased ubiquitylation activity of the 232-C protein in Figure 1/S1 compared to FL and RD?

We found that the 232-C protein batch used in the assay was not pure and have subsequently re-purified the protein. We have repeated the ubiquitination of ssDNA and RNA (Fig. 1H and 1I) and 232-C exhibited similar activity as WT. Furthermore, we performed autoubiquitination (Fig. S1G) and E2~Ub discharge assay (Fig. S1H) to compare the activity. 232-C was slower in autoubiquitination (Fig. S1G), but showed similar activity in the E2~Ub discharge assay as WT. These findings suggest that the RING domain in 232-C is functional and 232-C likely lacks ubiquitination site(s) present in 1-231 region necessary for autoubiquitination.

(3) Was it possible to positively identify the link between Ub and ssDNA/RNA using mass spectrometry? This would overcome issues associated with labels blocking binding rather than modification.

We have tried to use mass spectrometry to detect the linkage between Ub and ssDNA/RNA, but was unable to do so. We suspect that the oxyester linkage might be labile, posing a challenge for mass spectrometry techniques. Similarly, a recent preprint from Ahel lab, which utilises LC-MS, detects the Ub-NMP product rather than the linkage (https://www.biorxiv.org/content/10.1101/2024.04.19.590267v1.full.pdf).

(4) Furthermore, can a targeted MS approach be used to show that nucleotides are ubiquitylated in cells?

This will require future development and improvement of the MS approach, specifically the isolation of labile oxyester-linked products from cells and the optimisation of the MS detection method.

(5) Do the authors have the assignments (even partial?) for DTX3L RD? In Figure 4 it would be helpful to identify the peaks that correspond to the residues at the proposed binding site. Also do the shifts map to a defined surface or do they suggest an extended site, particularly for the ssDNA.

We only collected HSQC spectra which was insufficient for assignments. We have performed a competition experiment using ADPr and labelled ssDNA, showing that ADPr competes against the ubiquitination of ssDNA (Figure 4D). We have also provided an additional experiment showing that ssDNA with a blocked 3’-OH can compete against ubiquitination of ADPr (Figure 4E). These data, together with our NMR analysis, further strengthen the evidence that ssDNA and ADPr compete the same binding pocket in DTX3L RD. Understanding how DTX3L RD binds ssDNA/RNA is an ongoing research in the lab.

(6) Does sequence analysis help explain the specificity of activity for the family of proteins?

We have performed sequence alignment and structure comparison of DTX proteins using both RING and DTC domains (Fig. S3). These analyses showed that DTX3 and DTX3L RING domains lack a N-terminal helix and two loop insertions compared to DTX1, DTX2 and DTX4. These additions make DTX1, DTX2 and DTX4 RING domain larger than DTX3L and DTX3. It is not clear how these would influence the orientation of the recruited E2~Ub. Comparison of the DTC domain showed that DTX1, DTX2 and DTX4 contain an Ala-Arg motif, which causes a bulge at one end of DTC pocket. In the absence of Ala-Arg motif, DTC pockets of DTX3 and DTX3L contain an extended groove which might accommodate one or more of the nucleotides 5' to the targeted terminal nucleotide. It seems that both features of RING and DTC domains might attribute to the specificity of DTX3L and DTX3. We have included these comparisons in the discussion and suggested that future structural characterization is necessary to unveil the specificity.

(7) While including a summary mechanism (Figure 5I) is helpful, the schematic included does not necessarily make it easier for the reader to appreciate the key findings of the manuscript or to account for the specificity of activity observed. While this figure could be modified, it might also be helpful to highlight the range of substrates that DTX3L can modify - nucleotide, ADPr, ADPr on nucleotides etc.

We have modified this Figure to include the range of substrates.

**Reviewer #2 (Public Review):**
Summary:The manuscript by Dearlove et al. entitled "DTX3L ubiquitin ligase ubiquitinates single-stranded nucleic acids" reports a novel activity of a DELTEX E3 ligase family member, DTX3L, which can conjugate ubiquitin to the 3' hydroxyl of single-stranded oligonucleotides via an ester linkage. The findings that unmodified oligonucleotides can act as substrates for direct ubiquitylation and the identification of DTX3 as the enzyme capable of performing such oligonucleotide modification are novel, intriguing, and impactful because they represent a significant expansion of our view of the ubiquitin biology. The authors perform a detailed and diligent biochemical characterization of this novel activity, and key claims made in the article are well supported by experimental data. However, the studies leave room for some healthy skepticism about the physiological significance of the unique activity of DTX3 and DTX3L described by the authors because DTX3/DTX3L can also robustly attach ubiquitin to the ADP ribose moiety of NAD or ADP-ribosylated substrates. The study could be strengthened by a more direct and quantitative comparison between ubiquitylation of unmodified oligonucleotides by DTX3/DTX3L with the ubiquitylation of ADP-ribose, the activity that DTX3 and DTX3L share with the other members of the DELTEX family.Strengths:The manuscript reports a novel and exciting observation that ubiquitin can be directly attached to the 3' hydroxyl of unmodified, single-stranded oligonucleotides by DTX3L. The study builds on the extensive expertise and the impactful previous studies by the Huang laboratory of the DELTEX family of E3 ubiquitin ligases. The authors perform a detailed and diligent biochemical characterization of this novel activity, and all claims made in the article are well supported by experimental data. The manuscript is clearly written and easy to read, which further elevates the overall quality of submitted work. The findings are impactful and will help illuminate multiple avenues for future follow-up investigations that may help establish how this novel biochemical activity observed in vitro may contribute to the biological function of DTX3L. The authors demonstrate that the activity is unique to the DTX3/DTX3L members of the DELTEX family and show that the enzyme requires at least two single-stranded nucleotides at the 3' end of the oligonucleotide substrate and that the adenine nucleotide is preferred in the 3' position. Most notably, the authors describe a chimeric construct containing RING domain of DTX3L fused to the DTC domain DTX2, which displays robust NAD ubiquitylation, but lacks the ability to ubiquitylate unmodified oligonucleotides. This construct will be invaluable in the future cell-based studies of DTX3L biology that may help establish the physiological relevance of 3' ubiquitylation of nucleic acids.Weaknesses:The main weakness of the study is in the lack of direct evidence that the ubiquitylation of unmodified oligonucleotides reported by the authors plays any role in the biological function of DTX3L. The study leaves plenty of room for natural skepticism regarding the physiological relevance of the reported activity, because, akin to other DELTEX family members, DTX3 and DTX3L can also catalyze attachment of ubiquitin to NAD, ADP ribose and ADP-ribosylated substrates. Unfortunately, the study does not offer any quantitative comparison of the two distinct activities of the enzyme, which leaves plenty of room for doubt. One is left wondering, whether ubiquitylation of unmodified oligonucleotides is just a minor and artifactual side activity owing to the high concentration of the oligonucleotide substrates and E2~Ub conjugates present in the in-vitro conditions and the somewhat lower specificity of the DTX3 and DTX3L DTC domains (compared to DTX2 and other DELTEX family members) for ADP ribose over other adenine-containing substrates such as unmodified oligonucleotides, ADP/ATP/dADP/dATP, etc. The intriguing coincidence that DTX3L, which is the only DTX protein capable of ubiquitylating unmodified oligonucleotides, is also the only family member that contains nucleic acid interacting domains in the N-terminus, is suggestive but not compelling. A recently published DTX3L study by a competing laboratory (PMID: 38000390), which is not cited in the manuscript, suggests that ADP-ribose-modified nucleic acids could be the physiologically relevant substrates of DTX3L. That competing hypothesis appears more convincing than ubiquitylation of unmodified oligonucleotides because experiments in that study demonstrate that ubiquitylation of ADP-ribosylated oligos is quite robust in comparison to ubiquitylation of unmodified oligos, which is undetectable. It is possible that the unmodified oligonucleotides in the competing study did not have adenine in the 3' position, which may explain the apparent discrepancy between the two studies. In summary, a quantitative comparison of ubiquitylation of ADP ribose vs. unmodified oligonucleotides could strengthen the study.

We thank the reviewer for the constructive feedback. We agree that evidence for the biological function is lacking. While we have tried to detect Ub-ssDNA/RNA from cells, we found that isolating and detecting labile oxyester-linked Ub-ssDNA/RNA products remain challenging due to (1) low levels of Ub-ssDNA/RNA products, (2) the presence of DUBs and nucleases that rapidly remove the products during the experiments, and (3) our lack of a suitable MS approach to detect the product. For these reasons, we feel that discovering the biological function will require future effort and expertise and is beyond the scope of our current manuscript.

In the manuscript (PMID: 38000390), the authors used PARP10 to catalyse ADP-ribosylation onto 5’-phosphorylated ssDNA/RNA. They used the following sequences which lacks 3’-adenosine, which could explain the lack of ubiquitination.

E15_5′P_RNA [Phos]GUGGCGCGGAGACUU

E15_5′P_DNA [Phos]GTGGCGCGGAGACTT

We have performed the experiment using this sequence to verify this (see Author response image 2 below). We have cited this manuscript but for some reasons, Pubmed has updated its published date from mid 2023 to Jan 2024. We have updated the Endnote in the revised manuscript.

**Author response image 2. sa3fig2:** Fluorescently detected SDS-PAGE gel of *in vitro* ubiquitination catalysed by DTX3L-RD in the presence ubiquitination components and 6-FAM-labelled ssDNA D4 or D31.

We agree that it is crucial to compare ubiquitination of oligonucleotides and ADPr by DTX3L to find its preferred substrate. We have challenged oligonucleotide ubiquitination by adding excess ADPr and found that ADPr efficiently competes with oligonucleotide (Figure 4D). We have also performed an experiment showing that ssDNA with a blocked 3’-OH can compete against ubiquitination of ADPr (Figure 4E). These data support that ADPr and ssDNA compete for the same binding site on DTX3L.

We also performed kinetic analysis of ubiquitination of fluorescently labelled ssDNA (D4) and NAD+ by DTX3L-RD (Fig. 4F and Fig. S2D–G) to assess substrate preferences. Here, we used fluorescent-labelled NAD+ (F-NAD+) in place of ADPr as labelled NAD+ is commercially available. With the known concentration of fluorescently labelled ssDNA and NAD+ as the standard, we could estimate the rate of ubiquitinated product formation across different substrate concentrations. We have included this finding in the main text “DTX3L-RD displayed kcat value of 0.0358 ± 0.0034 min-1 and a Km value of 6.56 ± 1.80 μM for Ub-D4 formation, whereas the Michaelis-Menten curve did not reach saturation for Ub-F-NAD+ formation (Fig. 4F and fig. S2, D-G). Comparison of the estimated catalytic efficiency (kcat/Km = 5457 M-1 min-1 for D4 and estimated kcat/Km = 1190 M-1 min-1 for F-NAD+; Fig. 4F) suggested that DTX3L-RD exhibited 4.5-fold higher catalytic efficiency for D4 than F-NAD+. This difference primarily results from a better Km value for D4 compared to F-NAD+. Although DTX3L-RD showed weak Km for F-NAD+, it displays a higher rate for converting F-NAD+ to Ub-F-NAD+ at higher substrate concentration (Fig. 4F). Thus, substrate concentration will play a role in determining the preference.”

**Recommendations for the authors:**

**Reviewer #1 (Recommendations For The Authors):**
Writing/technical points:(1) The introduction is relatively complex and the last paragraph, which reviews the discoveries on the paper, is long. It may be helpful to highlight the significance and frame the experiments as what they have addressed, rather than detailing each set of experiments completed.

We have modified the last paragraph in the introduction to highlight the major discovery of our work.

(2) Line 24, Abstract. 'Its N-terminal region' is not obvious

We have changed “Its N-terminal region” to “the N-terminal region of DTX3L”.

(3) Line 44 - split sentence to emphasize E3 ligase point?

We have modified the sentence as suggested.

(4) Figures 1B and 1C could be larger - currently they are not very helpful. Also atoms (ADPr?) are shown, but not indicated in the legend or labelled on the panel.

We have enlarged Figures 1B and 1C and indicated RNA on the structure.

(5) The structure of the D2 domain of DTX3L has recently been reported (Vela-Rodriguez et al). It might be helpful to comment on this manuscript.

We have now commented on D2 domain in the results section and in the discussion.

(6) It would be helpful to indicate the DTX3L constructs used in Figure 1a.

We have included all DTX3L constructs used in Figure 1a.

(7) Interpretation of Figure 4A is difficult, the authors may wish to consider other ways to visualize the data.

We have now removed the black arrow in Figure 4A as it was confusing. Instead, we drew a black box on the cross-peak where the close-up views are shown in Figures 4B and 4C.

(8) Figure 4A. Please indicate which binding partner is highlighted by red/black arrows.

We have removed black arrow. The red arrows indicate cross-peaks which undergo chemical shift perturbation when DTX3L-RD was titrated with ssDNA or ADPr, highlighting their binding sites on DTX3L-RD overlap.

(9) Line 284 - please indicate the bulge in Figure S3.

We have indicated the bulge on Figure S3.

(10) Aspects of the discussion are speculative, given that evidence of Ub conjugated to nucleotides in cells is yet to be obtained and the functional consequences of modification are uncertain.

We understand that the discussion on the potential roles of ubiquitination of ssNAs is speculative. We have now modified it to: “Based on the known functions of the DTX3L/PARP9 complex and the findings of this study, we propose several hypotheses for future research”, so that readers will understand that these are speculative.

(11) Line 295 onwards - this paragraph discusses the role of the KH domains in nucleotide binding, but it is not clear that the authors have directly demonstrated that the KH domains bind nucleotides as all constructs used in the binding experiments in Figure 1/S1 include the RING-DTC domains.

We found that KH domains alone did not bind ssDNA or RNA. We have modified line 295. This section now reads “Typically, KH domains contain a GXXG motif within the loop between the first and second α helix (*22*). However, analysis of the sequence of the KHL domains in DTX3L shows these domains lack this motif. Multiple studies have shown that mutation in this motif abolishes binding to nucleic acids (*23-26*). Our findings show the DTX3L DTC domain binds nucleic acids but whether the KHL domains contribute to nucleic acid binding requires further investigation. Additionally, the structure of the first KHL domain was recently reported and shown to form a tetrameric assembly (*20*). Our analysis with DTX3L 232-C, which lacks the first KHL domain and RRM, indicate that it can still bind ssDNA and ssRNA. Despite this, a more detailed analysis will be required to determine whether oligomerization plays a role in nucleic acid binding and ubiquitination.”